# AN EFFICIENT GLOBAL-LOCAL FEATURE EXTRACTION ARCHITECTURE FOR 3D POINT CLOUDS

## ABSTRACT

Accurate 3D object detection and segmentation from LiDAR point clouds require both global context and fine-grained local features. Sparse convolutions capture local geometry efficiently but have limited receptive fields, while transformers model long-range context at high memory and runtime costs and often miss fine detail. We introduce Dilated Uniform Attention with 3D Sparse Convolution (DUA-SConv), a building block that integrates attention and sparse convolution in a complementary way. Each block applies self-attention over a uniformly dilated neighborhood spanning a large, fixed region to provide coarse global context, followed by sparse convolution to recover fine-grained local features. Stacked DUA-SConv blocks form a compact backbone that achieves high accuracy in 3D detection and segmentation with low runtime and parameter count.

## 1 INTRODUCTION

LiDAR point clouds provide rich spatial information, making them central to 3D detection. Modern detectors often adopt 3D sparse convolution backbones, which are computationally efficient with small kernels (e.g., 3×3×3) but limited in receptive field. Enlarging kernels improves global context but drastically increases cost and parameters. Recent methods Chen et al. (2023b); Feng et al. (2024); Lu et al. (2023) address this by efficiently expanding kernel sizes, surpassing standard sparse convolutions Yin et al. (2021), yet their receptive fields (up to 21×21×21) remain insufficient at fine voxel resolutions (e.g., 20 cm).

Transformers offer global context through self-attention over 3D voxels or points, but direct application is prohibitive due to the large number of points. Reducing complexity requires partitioning into smaller regions, which is challenging for sparse, irregular clouds. SST Fan et al. (2022a) and DSVT wan (2023) use structured partitioning, while Point Transformer V3 Wu et al. (2024) introduced point serialization with window-based self-attention, achieving strong performance but requiring about 4× more parameters than sparse convolution backbones Yin et al. (2021). Spatial state space models (recurrent neural networks) Zhang et al. (2024b); Liu et al. (2024) provide another means of capturing long-range dependencies, but their inherently sequential nature limits parallelism and slows runtime.

In this paper, we build on the efficiency and strong local feature extraction of 3D sparse convolutions, and propose a novel solution to their limited receptive field. The high-level concept of our method is illustrated in Fig. 1, column (c): each reference point attends to a dilated set of uniformly sampled points, capturing a large and consistent spatial region across distances. This enables efficient global context extraction at coarse resolution. After global features are computed, 3D sparse convolution is applied to the full-resolution (non-dilated) point cloud to propagate global context to nearby points and extract fine-grained local features. This structured integration of attention and convolution enables our backbone to capture both global and local context effectively and efficiently.

We propose the Dilated Uniform Attention with 3D Sparse Convolution (DUA-SConv) module to realize this concept, as illustrated in Fig. 2. DUA-SConv partitions the point cloud into uniformly dilated groups, applies efficient serialized window self-attention within each group, merges the updated features back to full resolution, and then applies 3D sparse convolution. Uniform dilation is key: it allows attention to span a large, fixed region with relatively few points, providing a consistent receptive field and geometric features across distance, thereby enabling efficient global feature learning. Stacked DUA-SConv blocks form a compact backbone that achieves strong accuracy in 3D

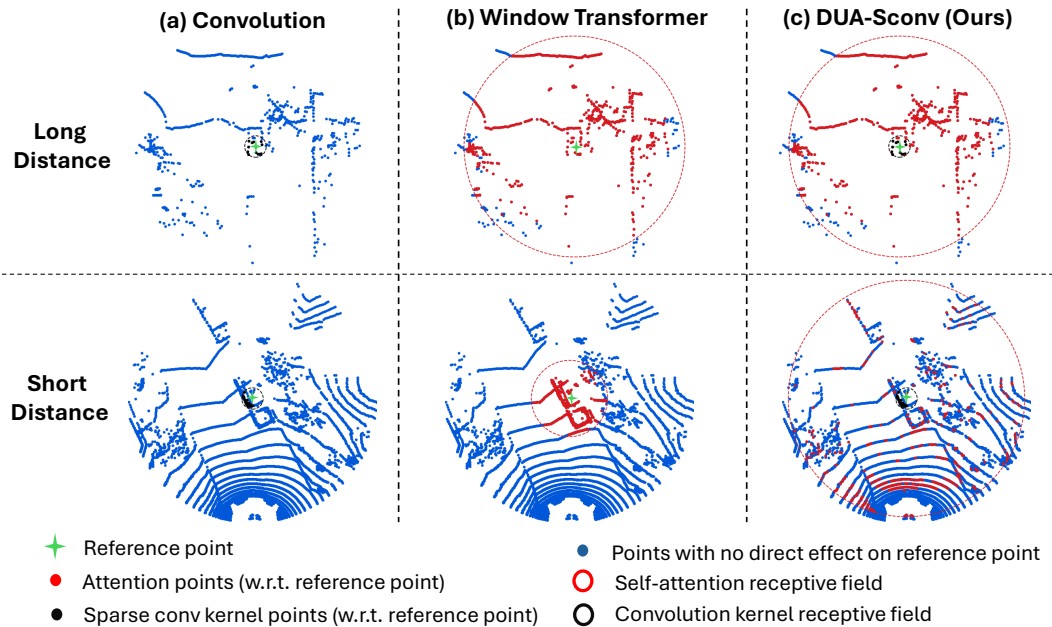

**(a) Convolution**  **(b) Window Transformer**  **(c) DUA-Sconv (Ours)**

Long Distance

Short Distance

+ Reference point
● Attention points (w.r.t. reference point)
● Sparse conv kernel points (w.r.t. reference point)
● Points with no direct effect on reference point
○ Self-attention receptive field
○ Convolution kernel receptive field

Figure 1: Comparison of three point cloud feature extraction methods (columns) on short-range (bottom) and long-range (top) scenes, with LiDAR point density naturally decreasing with distance. Each method's receptive region influencing the reference point (green star) is shown. The illustration is in bird's-eye view for clarity but represents 3D processing. (a) Convolution: Fixed-size kernel with limited receptive field and no access to global context. (b) Window Transformer: Self-attention over a fixed-size point window enables efficient parallelism, but varying density leads to inconsistent spatial coverage and overly small receptive fields at short range, limiting global feature extraction. (c) Ours: Self-attention with a fixed-size window over a dilated, uniformly sampled point cloud yields consistent, large receptive fields across distances. Global context is extracted at coarse resolution and refined by 3D sparse convolution, which shares context locally and captures fine details. This structured integration enables efficient and effective global-local feature extraction.

object detection and semantic segmentation while keeping parameter count and runtime low. The main contributions of this paper are:

- A novel DUA-SConv module that efficiently integrates global attention and local sparse convolution in a structured and complementary way.
- A uniform dilated grouping method that provides consistent coverage and geometry across varying densities, enabling efficient global context extraction with small attention windows.
- An efficient 3D global-local backbone built from stacked DUA-SConv blocks, achieving strong detection and segmentation performance with low parameter count and runtime.

## 2 RELATED WORK

**LiDAR-based 3D object detection and segmentation heads.** Modern 3D object detection and segmentation pipelines typically use a 3D backbone to produce dense bird's-eye view (BEV) features, which are then processed by detection or segmentation heads. For detection, CenterPoint Yin et al. (2021) applies a 2D CNN on BEV features to estimate object parameters, refined with point-level features, while SAFDNet Zhang et al. (2024a) enhances BEV features via adaptive diffusion. Transformer-based detection heads operate on BEV features and represent objects as queries Misra et al. (2021); Erabati & Araujo (2023); Zhou et al. (2022); Zhang et al. (2023a); Zhou et al. (2024); Bai et al. (2022), with FocalFormer3D Chen et al. (2023a) introducing multi-stage query refinement to better detect hard examples. For segmentation, heads typically process multi-scale 3D features Choy et al. (2019). This work focuses on improving the 3D backbone representation, offering a stronger feature extractor compatible with a wide range of existing detection and segmentation heads.

**3D sparse convolution backbones.** Standard 3D convolutions are computationally intensive for LiDAR point clouds due to sparsity and high dimensionality. Submanifold sparse convolutions Graham et al. (2018) improve efficiency by applying kernels only on non-empty voxels, becoming widely adopted in 3D detection backbones Yin et al. (2021); Deng et al. (2021); Zhang et al. (2024a); Wang et al. (2023); Chen et al. (2023c); Erabati & Araujo (2023); Zhang et al. (2023a); Zhou et al. (2024); Chen et al. (2023a). However, their receptive field is limited, and enlarging kernels causes cubic growth in computation. To mitigate this, HEDNet Zhang et al. (2023b) uses hierarchical sparse and dense encoder-decoder blocks. Other works propose efficient large-kernel designs: LargeKernel3D Chen et al. (2023b) reduces parameters via weight sharing, LSK3DNet Feng et al. (2024) prunes kernels with dynamic sparsity, and LinK Lu et al. (2023) generates kernels dynamically. Despite these advances, effective kernel sizes remain insufficient for capturing large objects or broad scene context, highlighting the need for larger receptive fields in sparse convolution frameworks.

**Attention based backbones.** Fully transformer-based 3D backbones have recently emerged. Since computing self-attention over all points (e.g., 100k points) is computationally prohibitive due to its quadratic complexity, the attention is restricted to local regions. However, region-based attention often relies on costly k-NN searches, which are inefficient in sparse point clouds. SST Fan et al. (2022a) avoids this by partitioning voxels into fixed BEV windows, but padding for parallelism reduces efficiency. FlatFormer Liu et al. (2023) flattens the point cloud and applies attention within window-sorted groups using alternating axes and shifts. DSVT wan (2023) partitions sparse voxels into alternating X/Y-axis windows for parallel attention. Point Transformer V3 Wu et al. (2024) serializes voxels into a 1D sequence based on proximity and applies windowed attention, achieving strong performance and fast runtime, but at the cost of 4× more parameters than sparse convolution backbones like CenterPoint Yin et al. (2021), and inconsistent receptive fields due to non-uniform density. This work addresses these limitations by introducing self-attention over uniformly sampled, dilated groups, achieving consistent spatial coverage and efficient global context extraction.

**Spherical Coordinate Approaches for Point Clouds.** Most LiDAR methods operate in Cartesian coordinates, where object shapes remain consistent across distances but point density decreases with range, complicating feature extraction. Cylinder3D Zhu et al. (2021) addresses range-dependent sparsity by transforming data into cylindrical coordinates, while CENet Cheng et al. (2022) projects points into 2D range images. These approaches reduce density variation but cause object size and shape to vary with distance, introducing new challenges for feature extraction. Sphereformer Lai et al. (2023) employs spherical coordinates, applying attention within frustum-shaped windows to enlarge receptive fields and improve information continuity. Unlike spherical-coordinate approaches, our method operates in Cartesian space, preserving object shape while achieving uniform density and large receptive fields through uniform dilated grouping with a transformer applied to each group.

**Point cloud self-supervised learning (SSL).** SSL for 3D point clouds reduces dependence on annotations and improves feature generalization. PointContrast Xie et al. (2020) explores scene-level contrastive learning, while MSC Wu et al. (2023) pushes beyond purely geometric cues via color and normal prediction. GroupContrast Wang et al. (2024) introduces graph-based segment guidance, and Sonata Wu et al. (2025) directly addresses the geometric shortcut and scales SSL pretraining to a large corpus, establishing a strong and reliable approach for point-cloud representation learning.

## 3 BACKGROUND - SERIALIZATION AND WINDOWED ATTENTION

As discussed in Sec. 2, global self-attention over large point clouds is computationally prohibitive, and mitigation strategies such as downscaling or k-NN-based local attention either lose detail or incur high computation overhead. A more efficient alternative serializes the point cloud into a 1D sequence that preserves spatial proximity Wu et al. (2024) and applies self-attention within fixed-size windows. This enables efficient batched processing but introduces new challenges. Because LiDAR point clouds are denser at close range and sparser at long range, fixed-size windows correspond to spatial regions of varying size and density, complicating feature learning and limiting generalization. Moreover, since attention complexity scales quadratically with window size, windows must remain small for efficiency, which restricts the receptive field in dense regions and weakens the ability to capture global context. Downscaling partly compensates by expanding the spatial extent of short windows, but at the cost of fine detail. This paper proposes a solution to these limitations of serialized, window-based self-attention in point cloud processing, as detailed in the next section.

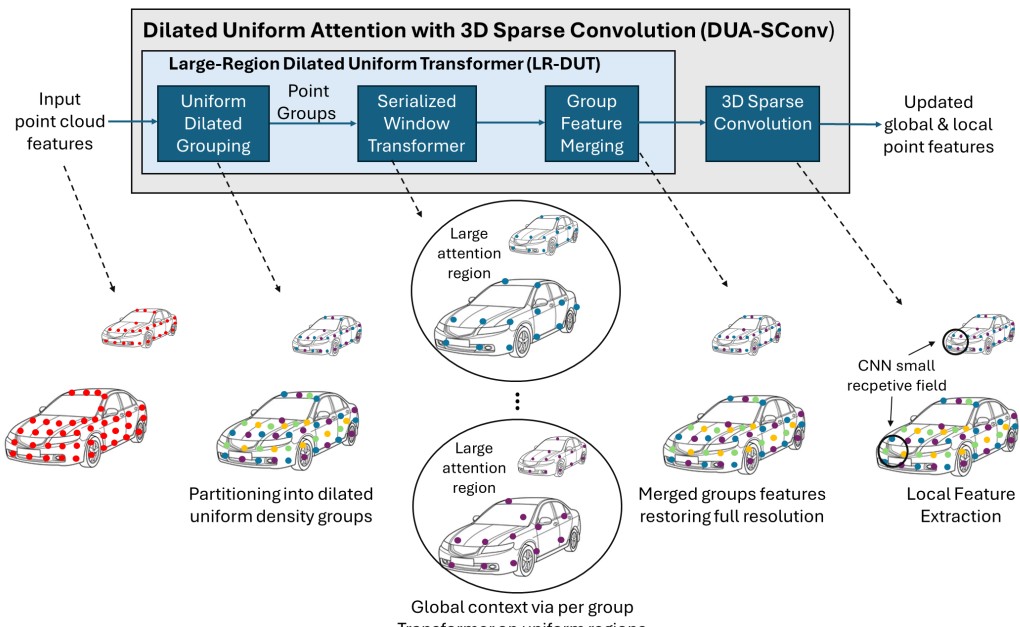

Figure 2: **Dilated Uniform Attention with 3D Sparse Convolution (DUA-SConv) module.** Global features are extracted by the Large-Region Dilated Uniform Transformer (LR-DUT), which partitions the input point cloud into groups of dilated points sampled with uniform spatial density across both range and groups. Each group is serialized into a 1D sequence based on spatial proximity, and self-attention is applied using fixed-size windows with shared weights across groups. This design ensures consistent spatial coverage and enables efficient global context learning with small, fixed-size attention windows. The resulting global features are then merged back to the full-resolution point cloud, where 3D sparse convolution propagates global context to nearby points and extracts fine-grained local features.

## 4 DILATED UNIFORM ATTENTION WITH 3D SPARSE CONVOLUTION

The global-local feature extraction concept in Fig. 1 is realized through our core architectural module, Dilated Uniform Attention with 3D Sparse Convolution (DUA-SConv), shown in Fig. 2. DUA-SConv combines a Large-Region Dilated Uniform Transformer (LR-DUT) with a 3D sparse convolution layer. LR-DUT partitions the point cloud into uniformly dense, dilated groups, applies serialized windowed self-attention within each group, and merges the updated features back into the full-resolution point cloud. Window attention over the dilated, uniformly sampled points enables each point to attend over a large and consistent spatial region using a small window size, ensuring efficiency and stable context across distances. The resulting global features are coarse due to the use of dilated points and are then refined by sparse convolution applied to the full-resolution point cloud, which diffuses global context among neighboring points and captures fine-grained local geometry. This structured integration of global attention and local convolution forms the backbone of our architecture, detailed in subsection 4.3. The following sections outline the DUA-SConv components.

### 4.1 LARGE-REGION DILATED UNIFORM TRANSFORMER (LR-DUT)

The LR-DUT consists of three main components, as illustrated in Fig. 2: Uniform Dilated Grouping, Serialized Window Transformer, and Group Feature Merging, described in the following.

**Uniform Dilated Grouping (UDG).** UDG addresses the challenges of applying window-based self-attention to non-uniform point densities, as discussed in Sec. 3, by partitioning the point cloud into multiple groups, each containing a uniformly dilated subset of points with consistent spatial density across distance and across groups. This enables fixed-size window self-attention to operate within each group without being affected by density variations that typically hinder performance.

LiDAR point cloud density decreases quadratically with distance due to the sensor's spherical coordinate emission geometry. This non-uniform density can be converted to a uniform density by applying a range-dependent dilation factor, defined as $\mu(R) = (R_{\max}/R)^2$, where $R$ is the point's range and $R_{\max}$ is the maximum LiDAR range. This results in stronger dilation at close range and no dilation at the maximum range, compensating for the quadratic drop in density and equalizing the point cloud density across distances. To implement this, the spherical coordinates of each point are computed and quantized into range, azimuth, and elevation bins, with spacing in each dimension no greater than the sensor's native resolution. Dilation is applied independently in azimuth and elevation using a per-dimension factor of $\sqrt{\mu(R)} = R_{max}/R$. Since non-integer dilation would require interpolation, which is inefficient and error-prone, we approximate the dilation using integer values: $\tilde{\mu}(R) = \lfloor R_{max}/R \rfloor^2$, where $\lfloor \cdot \rfloor$ denotes rounding down to the nearest integer. This results in a piecewise-constant dilation across range intervals. We found that using four intervals provides a good balance between performance and simplicity, and adopt the fixed intervals given in Eq. 1.

$$
\tilde{\mu}(R) = \begin{cases} 1 & \text{if } \frac{R_{\max}}{2} < R \le R_{\max} \\ 4 & \text{if } \frac{R_{\max}}{4} < R \le \frac{R_{\max}}{2} \\ 16 & \text{if } \frac{R_{\max}}{8} < R \le \frac{R_{\max}}{4} \\ 64 & \text{if } 0 < R \le \frac{R_{\max}}{8} \end{cases} \quad (1)
$$

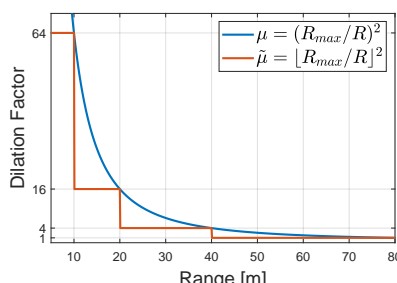

Figure 3: Dilation factor $\mu(R)$ vs. $\tilde{\mu}(R)$.

Fig. 3 visualizes both the continuous dilation function $\mu(R)$ and its piecewise approximation $\tilde{\mu}(R)$ using these four intervals. We observed that adding more intervals provided no noticeable accuracy gain and increased implementation complexity.

Based on the piecewise dilation intervals, points within each range band are partitioned into groups with approximately uniform density, as illustrated in Fig. 4. The figure illustrates the four range intervals and the associated number of groups per interval, as specified in equation 1. For a given interval with dilation factor $\tilde{\mu}(R) = N$, we perform uniform grid-based downsampling in both azimuth and elevation by a factor of $\sqrt{N}$, resulting in $N$ groups generated using different grid phase offsets. The azimuth and elevation grids of these groups are visualized in different colors and labeled by their corresponding indices in Fig. 4, and are formally defined as follows. Let $\theta[i] = i\Delta_\theta$ and $\phi[j] = j\Delta_\phi$ denote the azimuth and elevation grid values, where $\Delta_\theta$ and $\Delta_\phi$ are the respective grid resolutions, and $i = 1, \ldots, I$, $j = 1, \ldots, J$ are the grid indices. Similarly, let $r[m] = m\Delta_R$ represent the range grid values, where $\Delta_R$ is the range grid spacing and $m = 1, \ldots, M$ is the range index. For a given range interval with dilation factor $N$, there are $N$ groups within the range band $\frac{R_{\max}}{2\sqrt{N}} < R \le \frac{R_{\max}}{\sqrt{N}}$, as defined in equation 1. The voxels in each group, $\Omega_{k,q}^N$, are defined by a unique pair of grid phase offsets $(k,q) \in \{1, \ldots, \sqrt{N}\}^2$, and are given by:

$$
\Omega_{k,q}^N = \{ r[m], \ \theta[k + u\sqrt{N}], \ \phi[q + v\sqrt{N}] \ \text{for} \tag{2}
$$

$$
m = \frac{M}{2\sqrt{N}}, \ldots, \frac{M}{\sqrt{N}}, \ u = 0, \ldots, \left\lfloor \frac{I}{\sqrt{N}} \right\rfloor, \ v = 0, \ldots, \left\lfloor \frac{J}{\sqrt{N}} \right\rfloor \}. \tag{3}
$$

This construction yields $N$ uniformly dilated groups per range interval, each corresponding to a distinct grid phase offset, as illustrated in Fig. 4. The final output of the UDG block is the union of all $\Omega_{k,q}^N$ groups for $(k,q) \in \{1, \ldots, \sqrt{N}\}^2$, across all range intervals with dilation factors $N \in \{1, 4, 16, 64\}$. Note that dilation is based on spherical coordinates, while the point cloud itself remains in Cartesian space. In this space, each group maintains approximately uniform density and object shape across distance, providing consistent geometric features and a consistent receptive field for each attention window, both aiding feature extraction. This differs from spherical point cloud representation methods (see Sec. 2), which enforce equal spacing in spherical coordinates but produce inconsistent object shapes across distance.

In practice, the spatial extent of each group slightly exceeds the strict boundaries defined in Eq. 1. This ensures that groups include points near the edges of adjacent range intervals, providing smoother

transitions and allowing each group's self-attention window to incorporate neighboring context. These small overlaps help avoid boundary artifacts and ensure that attention is not restricted to artificially isolated bands. See the Supplementary Material for more details.

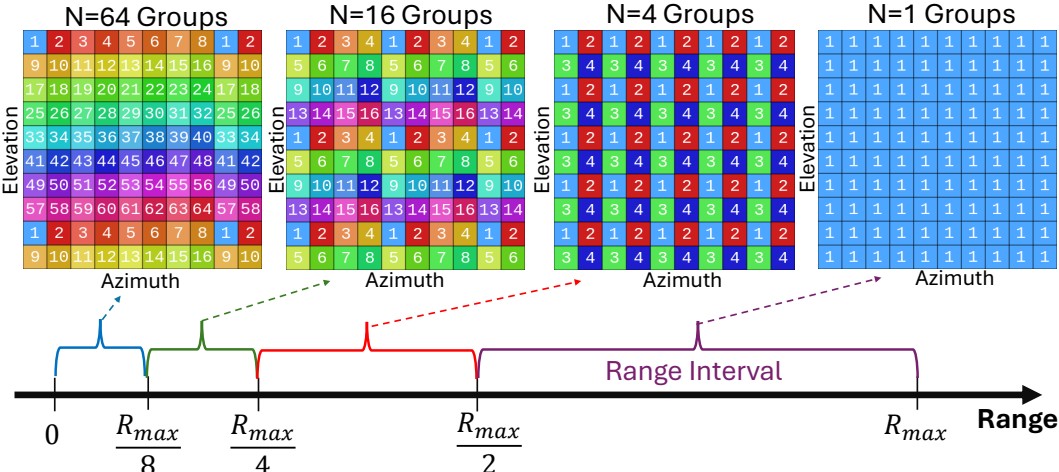

Figure 4: **Uniform Dilated Grouping (UDG)** partitions the point cloud into groups with approximately uniform density. This is achieved by dividing the point cloud into four range intervals (defined in equation 1), each marked with a different color along the range dimension. The number of groups per interval decreases with range, $N = 64, 16, 4, 1$, resulting in stronger dilation at short range and no dilation at long range. Within each interval, groups are formed by downsampling the azimuth and elevation grids, and each group is visualized using a distinct color and index.

**Large and Uniform Region Transformer.** After UDG partitions the point cloud into uniformly dense, dilated groups $\Omega_{k,q}^N$, each group is serialized into a 1D sequence following Wu et al. (2024), with varying scanning patterns across groups to enhance representation diversity. A shared-weight multi-layer Transformer is then applied independently to each group. For each point in the serialized sequence, attention is computed only with its $K$ immediate neighbors using a fixed-size sliding window. This point-wise attention maintains linear complexity with respect to the number of points, while allowing each point to capture global context over a consistent and large spatial extent with a relatively small window size $K$, enabled by the uniform and dilated density introduced by UDG.

To encode spatial relationships, we add positional information to the keys and queries. A key challenge in window-based attention is representing relative positions effectively Wu et al. (2024), as explicitly computing the offset between each query and its attended points is computationally expensive and slows down inference. To overcome this, we use the absolute 3D coordinates $(x, y, z)$ of each point as positional input. These coordinates are passed through a lightweight, one-layer MLP to produce high-dimensional embeddings, which are added to the keys and queries. This enables the Transformer to implicitly model relative positions through dot-product attention, as the difference in absolute positions influences the angle and magnitude between the embeddings, capturing spatial relationships without the overhead of explicit offset computation. Further details on the relative positional encoding and the transformer module are given in the Supplementary Material.

**Group Feature Merging.** After the Transformer is applied to each dilated group $\Omega_{k,q}^N$, the updated features are merged back to the original point cloud resolution by taking the union of all groups across all range intervals. This yields point-level features aligned with the input grid, where each point now incorporates global contextual information from a large receptive field. This completes the global feature extraction stage. Local geometric features are then extracted via 3D sparse convolution on the merged point cloud, as described in subsection 4.2.

**Enhancing Training via Group-wise Dropout.** To reduce training time we introduce a group-level dropout mechanism. Specifically, groups are randomly selected, based on a predefined dropout probability, to bypass the transformer stage. As a result, points in dropped groups do not receive global context directly. However, neighboring points in retained groups do obtain global features,

which are subsequently propagated through the sparse convolution, enabling global context sharing. This approach reduces the number of transformer operations, lowering computational cost while preserving all points and maintaining fine-grained geometric details, unlike conventional downscaling or pooling methods that reduce point resolution during training and inference Wu et al. (2024).

### 4.2 3D SPARSE CONVOLUTION

The LR-DUT block in subsection 4.1 updates every point in the full-resolution cloud with global contextual features extracted from a large region of dilated sampled points. These updated full-resolution points are voxelized in a Cartesian grid and processed using 3D submanifold sparse convolution Graham et al. (2018) with a small kernel size of $3 \times 3 \times 3$. This convolution efficiently propagates coarse global context across fine-resolution neighborhoods and extracts local features.

### 4.3 DUA-SCONV LAYERED BACKBONE

Our 3D feature extraction backbone consists of a cascade of DUA-SConv blocks, as shown in Fig. 5. The input point cloud is voxelized into a Cartesian grid and processed by a Stem block Yin et al. (2021) to generate initial features, followed by four sequential DUA-SConv blocks. Each block downsamples spatial resolution by a factor of 2 in all dimensions, and its transformer module contains four self-attention layers. We use four DUA-SConv blocks to match established backbones such as CenterPoint Yin et al. (2021) and LinK Lu et al. (2023), with convolution layers identical to CenterPoint for fair runtime and parameter comparisons. For object detection, final-stage features $(f_4)$ are passed to a detection head (as in CenterPoint), while for semantic segmentation, multi-scale features from all stages $(f_1, f_2, f_3, f_4)$ are aggregated and processed by a segmentation head, following designs such as MinkowskiNet Choy et al. (2019).

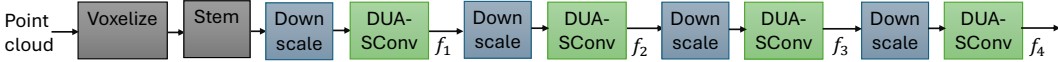

Figure 5: **DUA-SConv Based Backbone.** Detection head uses only final-stage features $(f_4)$, while segmentation head uses multi-scale features $(f_1 - f_4)$.

## 5 RESULTS

**Implementation details.** Information on the proposed backbone's implementation, training, inference, and the performance evaluation is provided in the Supplementary Material.

**Datasets.** We evaluate 3D detection on nuScenes Caesar et al. (2020) and Waymo Sun et al. (2020), and 3D semantic segmentation on SemanticKITTI Behley et al. (2019), nuScenes, Waymo, ScanNet Dai et al. (2017), and S3DIS Armeni et al. (2016). Standard metrics are used for each benchmark.

**Reference methods** We evaluate the impact of replacing existing backbones with ours for 3D object detection and segmentation. For detection, we evaluate our backbone with both the CenterPoint and the stronger FocalFormer3D heads Chen et al. (2023a). For segmentation, we follow LinK and LargeKernel3D and adopt the standard MinkowskiNet head Choy et al. (2019).

### 5.1 3D OBJECT DETECTION

Table 1 reports 3D detection results on the nuScenes validation and test sets, and Table 2 shows LEVEL-2 results on the Waymo validation set. Class-wise and LEVEL-1 results on Waymo are provided in the Supplementary Material. We denote our backbone with CenterPoint as *Ours (CenterPoint)* and with FocalFormer3D as *Ours (F. Former)*. We evaluate two backbone sizes: a base model (*Ours-B*) and a scaled-up version (*Ours-L*), which doubles the Transformer feature dimension and attention heads, increasing parameters from 10.8M to 16.2M and aligning its capacity with leading object detection methods. Implementation details appear in the Appendix E, and size/runtime analysis in subsection 5.4. All results are reported without test-time augmentation, as methods differ in their use of geometric transforms, instance-level augmentation, and model ensembling, making fair comparison difficult. For the nuScenes test server, we submitted only the FocalFormer3D variant with the base model size due to the limit on the number of allowed submissions.

Table 1: nuScenes *val* and *test* sets. "–" = not reported. Best per column in **bold**.

| Methods | Val. | | Test | | | | | | | | | | | |
|---|---|---|---|---|---|---|---|---|---|---|---|---|---|---|
| | mAP | NDS | mAP | NDS | Car | Trk | C.V. | Bus | Trl | Bar | Mot | Bike | Ped | T.C. |
| CenterPoint | 59.0 | 66.4 | 58.0 | 65.5 | 84.6 | 51.0 | 17.5 | 60.2 | 53.2 | 70.9 | 53.7 | 28.7 | 83.4 | 76.7 |
| TransFusion-L | 60.0 | 66.8 | 65.5 | 70.2 | 86.2 | 56.7 | 28.2 | 66.3 | 58.8 | **78.2** | 68.3 | 44.2 | 86.1 | 82.0 |
| Focals Conv | 61.2 | 68.1 | 63.8 | 70.0 | 86.7 | 56.3 | 23.8 | 67.7 | 59.5 | 74.1 | 64.5 | 36.3 | 87.5 | 81.4 |
| SphereFormer | – | – | 65.5 | 70.7 | 84.9 | 55.1 | 29.9 | 66.4 | 59.3 | 75.2 | 71.4 | 47.1 | 86.0 | 79.7 |
| LargeKernel3D | 63.3 | 69.1 | 65.3 | 70.5 | 85.9 | 55.3 | 26.8 | 66.2 | 60.2 | 74.3 | 72.5 | 46.6 | 85.6 | 80.0 |
| LinK | 63.6 | 69.5 | 66.3 | 71.0 | 86.1 | 55.7 | 30.9 | 65.7 | 62.1 | 75.5 | 73.5 | 47.5 | 85.8 | 80.4 |
| HEDNet | 66.7 | 71.4 | 67.7 | 72.0 | 87.1 | 56.5 | 33.6 | 70.4 | 63.5 | 78.1 | 70.4 | 44.8 | 87.9 | 85.1 |
| ScatterFormer | 68.3 | 72.4 | – | – | – | – | – | – | – | – | – | – | – | – |
| FSHNet | 68.1 | 71.7 | – | – | – | – | – | – | – | – | – | – | – | – |
| LION | 68.0 | 72.1 | 69.8 | 73.9 | 87.2 | **61.1** | 36.3 | 68.9 | 65.0 | **79.5** | 74.0 | 49.2 | 90.0 | **87.3** |
| UniMamba | 68.5 | 72.6 | **70.2** | **74.0** | **87.9** | 60.4 | 36.7 | **70.9** | 65.9 | 79.4 | 73.5 | 49.5 | **90.5** | 86.9 |
| DSVT | 66.4 | 71.1 | 68.4 | 72.7 | – | – | – | – | – | – | – | – | – | – |
| Voxel Mamba | 67.5 | 71.9 | 69.0 | 73.0 | 86.8 | 57.1 | 35.4 | 68.0 | 63.2 | 77.3 | 74.7 | **50.8** | 89.5 | 86.9 |
| SAFDNet | 66.3 | 71.0 | 68.3 | 72.3 | 87.3 | 57.3 | 37.3 | 68.0 | 63.7 | **79.5** | 71.1 | 44.8 | 89.0 | 84.9 |
| FocalFormer3D | 66.5 | 71.1 | 68.7 | 72.6 | 87.2 | 57.1 | 34.4 | 69.6 | **64.9** | 77.8 | 76.2 | 49.6 | 88.2 | 82.3 |
| Ours-B (CenterPoint) | 65.4 | 70.6 | – | – | – | – | – | – | – | – | – | – | – | – |
| Ours-B (F. Former) | 67.8 | 72.0 | 70.1 | 73.3 | 87.7 | 57.8 | **38.5** | 70.6 | 64.7 | 78.1 | **78.7** | 49.8 | 89.9 | 85.5 |
| Ours-L (F. Former) | **68.9** | **72.8** | – | – | – | – | – | – | – | – | – | – | – | – |

Table 2: Waymo validation (LEVEL-2, single frame).

| Methods | mAP | mAPH |
|---|---|---|
| CenterPoint | 69.8 | 67.6 |
| DSVT | 74.0 | 72.1 |
| HEDNet | 75.3 | 73.4 |
| ScatterFormer | – | 73.8 |
| FSHNet | **77.1** | 74.9 |
| PTV3 | – | 73.0 |
| LION | 75.1 | 73.2 |
| UniMamba | 76.1 | 74.1 |
| Voxel Mamba | – | 73.6 |
| SAFDNet | 75.7 | 73.9 |
| FocalFormer3D | 71.5 | 69.0 |
| Ours-B (CenterPoint) | 70.9 | 68.8 |
| Ours-B (F. Former) | 75.9 | 74.2 |
| Ours-L (F. Former) | 77.0 | **75.1** |

Table 3: Segmentation mIoU on validation sets. nS = nuScenes, SemK = SemanticKITTI, Way = Waymo, SN = ScanNet, S3 = S3DIS Area 5, #Prm = parameters [M].

| Methods | nS | SemK | Way | SN | S3 | #Prm |
|---|---|---|---|---|---|---|
| PointNeXt | – | – | – | 71.5 | 70.5 | 41.6 |
| Swin3D | – | – | – | 76.4 | 72.5 | 23.5 |
| ST | – | – | – | 74.3 | 72.0 | 18.8 |
| Cylinder3D | 76.1 | 64.3 | – | – | – | 55.9 |
| LSK3DNet | 80.1 | – | – | 75.7 | – | 28.8 |
| SphereFormer | 78.4 | 67.8 | 69.9 | – | – | 32.3 |
| PTV3 | 80.4 | 70.8 | 71.3 | **77.5** | **73.4** | 46.2 |
| Ours-H (Mink) | **80.7** | **71.0** | **71.7** | 76.6 | 72.6 | 44.7 |
| MinkUNet | 73.3 | 63.8 | 65.9 | 72.2 | 65.4 | 8.5 |
| SPVNAS | 77.4 | 64.7 | – | – | – | 12.5 |
| LinK | – | 67.5 | – | – | – | 10.7 |
| Ours-B (Mink) | 78.1 | 67.9 | 69.5 | 75.8 | 72.2 | 10.8 |

Our base size backbone paired with the CenterPoint head outperforms CenterPoint, LinK, and LargeKernel3D on both nuScenes and Waymo, all using the same detection head, highlighting the strength of our design. Using our backbone in FocalFormer3D further improves results, outperforming prior methods on both datasets. Our method with the baseline model size ranks highly on the nuScenes leaderboard without test-time augmentation or camera fusion. Scaling-up the model further improves accuracy, achieving state-of-the-art performance on nuScenes and leading in mAPH on Waymo, while adding only moderate increases in parameters and latency (see subsection 5.4).

## 5.2 3D SEMANTIC SEGMENTATION

Table 3 presents 3D semantic segmentation results with model sizes. Unlike detection, segmentation results are typically reported only with test-time augmentation (TTA), and TTA strategies vary widely and are often under-documented, making fair comparison difficult. We therefore report segmentation results on validation sets without TTA rather than test-set results. We pair our backbone with the MinkUNet head Choy et al. (2019) for segmentation. We evaluate two variants: a base model (*Ours-B (Mink)*) and a scaled-up model (*Ours-H (Mink)*), which increases the Transformer feature dimension

Table 4: Ablation on *nuScenes* val.

| ID | Win. Tr. | Group | Win. Size | Pos. Enc. | mAP | NDS |
|----|----------|-------|-----------|-----------|------|------|
| 1 | No | – | – | - | 59.0 | 66.4 |
| 2 | Yes | None | 10K | K,Q | 60.8 | 66.9 |
| 3 | Yes | None | 1K | K,Q | 61.4 | 67.3 |
| 4 | Yes | Rand. | 1K | K,Q | 62.6 | 67.6 |
| 5 | Yes | FPS | 1K | K,Q | 63.2 | 68.0 |
| 6 | Yes | UDG | 1K | None | 64.0 | 69.6 |
| 7 | Yes | UDG | 1K | K,Q,V | 64.9 | 70.1 |
| 8 | Yes | UDG | 1K | K,Q | 65.3 | 70.3 |

Table 5: Model size & runtime on nuScenes validation.

| Methods | # Parm [M]. | Time [ms] | mAP | NDS |
|---------|-------------|-----------|------|------|
| LinK | 10.3 | 109 | 63.6 | 69.5 |
| HEDNet | 15.3 | 67 | 66.7 | 71.4 |
| ScatterFormer | 12.6 | 45 | 68.3 | 72.4 |
| LION | 16.1 | 152 | 68.0 | 72.1 |
| UniMamba | 16.3 | 121 | 68.5 | 72.6 |
| Voxel Mamba | 15.1 | 182 | 67.5 | 71.9 |
| FSHNet | 13.1 | 123 | 68.1 | 71.7 |
| SAFDNet | 15.8 | 106 | 66.3 | 71.0 |
| Ours (F.Former) | 10.8 | 119 | 67.8 | 72.0 |
| Ours-L (F.Former) | 16.9 | 137 | **68.9** | **72.8** |

by a factor of 3 and doubles the number of layers (details in Appendix E), raising the parameter count from 10.8M to 44.7M and aligning its capacity with leading segmentation methods. Methods in Table 3 are grouped by capacity (separated by a horizontal line): the upper block compares large models to *Ours-H (Mink)*, and the lower block compares smaller models to *Ours-B (Mink)*.

In the lower-capacity group, replacing the original MinkUNet backbone with our base model yields a clear performance gain with only a modest increase in parameters. Scaling up the backbone further improves results, outperforming all comparable-size models on long-range outdoor datasets (nuScenes, KITTI, Waymo) and approaching, though still below, PTV3 performance on short-range indoor datasets (ScanNet, S3DIS). This suggests that our range-normalized design is particularly effective when point-density variation with distance is more pronounced, as in outdoor LiDAR scenes.

## 5.3 ABLATION STUDY

Table 4 shows an ablation study on the nuScenes validation set using the CenterPoint head. Row (1) is the baseline backbone without the LR-DUT module in DUA-SConv; row (8) is the full DUA-SConv. Rows (2)–(7) evaluate variations of the LR-DUT module, focusing on: (a) Grouping strategies: random sampling, consecutive farthest point sampling, UDG, or no grouping; (b) Transformer window size: 1K vs. 10K, matched in average spatial coverage to 1K without UDG; (c) Positional encoding: applied to all tokens vs. only to keys and queries. Results show that adding transformer layers to sparse convolution yields moderate gains. Grouping consistently improves performance, with UDG clearly outperforming alternatives. Increasing the window size alone, without point dilation, does not help. Positional encoding is beneficial, especially when applied only to keys and queries, likely due to improved relative positioning.

## 5.4 RUNTIME AND MODEL SIZE EVALUATION

Table 5 compares parameter count, runtime, and performance on nuScenes for our detection backbone and reference methods (runtime measured on a single RTX 3090 GPU). Our base model achieves strong performance with a relatively small parameter count and low runtime, while the scaled variant further improves accuracy with only a modest increase in model size and runtime. This configuration reaches state-of-the-art performance with a small overhead in model size and runtime compared to top-performing backbones such as UniMamba Jin et al. (2025) and FSHNet Liu et al. (2025), though it is notably slower than ScatterFormer He et al. (2024) and HEDNet Zhang et al. (2023b).

## 6 CONCLUSION

We introduced a compact 3D backbone for point cloud processing that efficiently captures both global and local context. At its core is the DUA-SConv module, which combines serialized window self-attention with 3D sparse convolution in a structured, complementary design. Global features are extracted from uniformly sampled, range-dilated point groups, ensuring consistent spatial coverage across distances. These features are then refined with sparse convolution to capture local geometry. Stacking DUA-SConv blocks results in a backbone with modest model size and runtime, while achieving superior detection and segmentation performance compared to similarly sized backbones.

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

# A APPENDIX

This appendix includes supplementary details on implementation, additional results, and extended analyses that support and complement the main paper.

# B DETAILS ON ATTENTION IMPLEMENTATION

The transformer block used in the LR-DUT module is illustrated in Fig. 6. A sliding window is applied over the serialized sequence of points within each group. The window moves one point at a time, and at each step, the point at the center of the window is designated as the query. All points within the window act as keys and values. The attention output is computed only for the center (query) point, which is then updated. This process is repeated as the window slides across the sequence, ensuring that each point is updated using its window context while maintaining a consistent window size. To support this localized attention, we incorporate an efficient form of relative positional encoding, as described next.

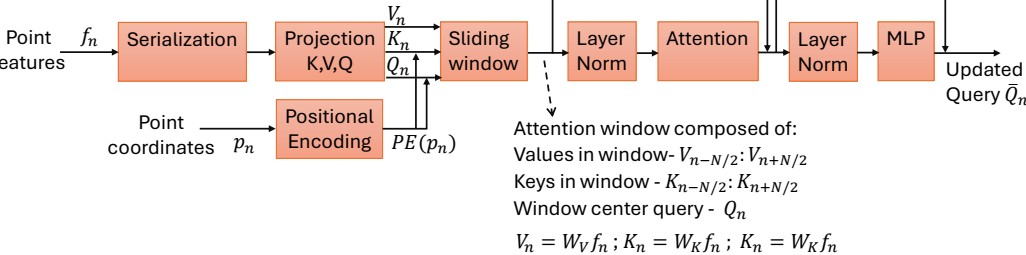

Figure 6: Serialized window attention implementation in LR-DUT

## EFFICIENT RELATIVE POSITIONAL ENCODING VIA KEYS AND QUERIES

Positional encoding in Transformers is typically implemented by adding absolute position information to the queries, keys, and values. While this approach is computationally efficient, it has been shown that relative positional encoding is more effective for capturing local geometric relationships in self-attention Lai et al. (2022); Yang et al.; Zhao et al. (2021). However, computing explicit relative positions between every query and key point introduces significant overhead Wu et al. (2024), especially in 3D point clouds.

To overcome this limitation, we adopt an efficient approximation: we add absolute positional encoding only to the queries and keys, not to the values. Each point's 3D coordinates $(x, y, z)$ are mapped to the feature space via a one-layer MLP. This design enables the attention mechanism to capture relative spatial relationships through the interaction of position-encoded queries and keys, while keeping the value features purely content-based.

Let each point $i$ have coordinates $\mathbf{p}_i = (x_i, y_i, z_i)$ and a feature vector $\mathbf{f}_i$. We define:

$$
\begin{aligned}
Q_i &= W_Q \mathbf{f}_i + \mathrm{PE}(\mathbf{p}_i), \\
K_j &= W_K \mathbf{f}_j + \mathrm{PE}(\mathbf{p}_j), \\
V_j &= W_V \mathbf{f}_j,
\end{aligned}
$$

where $W_Q$, $W_K$, and $W_V$ are learnable projection matrices, and $\mathrm{PE}(\cdot)$ is a positional encoding function (an MLP that maps 3D coordinates to the feature space). Here, $Q_i$ denotes the query for the $i$-th point, and $K_j$ and $V_j$ are the key and value for the $j$-th point.

The attention score between query point $i$ and key point $j$, both of dimension $d$, is given by:

$$
\alpha_{ij} \propto \exp\left( \frac{Q_i^\top K_j}{\sqrt{d}} \right),
$$

which expands to:

$$
\alpha_{ij} \propto \exp\left( \frac{(W_Q \mathbf{f}_i)^\top (W_K \mathbf{f}_j) + (W_Q \mathbf{f}_i)^\top \mathrm{PE}(\mathbf{p}_j) + \mathrm{PE}(\mathbf{p}_i)^\top (W_K \mathbf{f}_j) + \mathrm{PE}(\mathbf{p}_i)^\top \mathrm{PE}(\mathbf{p}_j)}{\sqrt{d}} \right).
$$

The final term, $\mathrm{PE}(\mathbf{p}_i)^\top \mathrm{PE}(\mathbf{p}_j)$, captures the geometric relationship between the positions of the query and key points, enabling the attention weights to encode relative spatial structure. Although $\mathrm{PE}(\cdot)$ encodes absolute positions, the dot product between the encoded positions reflects how similar their transformed representations are in feature space, implicitly encoding spatial proximity or alignment. Since this interaction is symmetric and depends on the positional relationship between points, it provides a learned notion of relative position.

Given the attention weights $\alpha_{ij}$, the output feature for the $i$-th query point is computed as a weighted sum of the values from its attention window:

$$
\mathbf{z}_i = \sum_{j \in \mathcal{N}(i)} \alpha_{ij} V_j,
$$

where $\mathcal{N}(i)$ denotes the set of key points within the attention window centered at point $i$, and $V_j = W_V \mathbf{f}_j$ is the value vector for point $j$. Since positional encoding is not applied to $V_j$, the output feature $\mathbf{z}_i$ is purely a content-based aggregation, modulated by spatially-aware attention weights that reflect the positional relationship between the query and key points.

This mechanism is analogous to convolution, where each kernel weight is associated with a fixed relative offset from the center. Similarly, in our case, the attention weight $\alpha_{ij}$ depends on the spatial offset between the query (center) point and its neighboring key points, playing a similar role in spatially structured feature aggregation. In both cases, the aggregated features themselves remain position-agnostic; the spatial structure is captured solely by the coefficients used in the aggregation - kernel weights in convolution, and attention weights in our method.

## C  HANDLING ATTENTION AT RANGE INTERVAL BOUNDARIES

In Section 4.1 of the main paper, we describe how the LR-DUT module partitions the point cloud into range intervals (bands), and further into groups within each interval. Each group is serialized, and window-based self-attention is applied to the sequence of points in that group.

A challenge arises at the boundaries of range intervals: points near the edge of an interval lack neighbors on one side, as nearby points in that direction belong to an adjacent interval. This breaks the assumption of symmetric context in window-based attention and can reduce its effectiveness.

To address this, we extend each group by including additional points from the neighboring range interval - specifically, up to half the attention window size. These extended points are dilated using

Table 6: DUA-SConv hyperparameter optimization on the nuScenes validation set with the Focal-Former3D detection head (F. Former).

| $K$ | $L$ | $N$ | $P_d$ | mAP |
|---|---|---|---|---|
| 256 | 4 | 4 | 0.5 | 66.2 |
| 512 | 4 | 4 | 0.5 | 66.6 |
| 1024 | 4 | 4 | 0.5 | **67.8** |
| 2048 | 4 | 4 | 0.5 | 67.5 |
| 1024 | 2 | 4 | 0.5 | 67.0 |
| 1024 | 3 | 4 | 0.5 | 67.4 |
| 1024 | 5 | 4 | 0.5 | **67.8** |
| 1024 | 4 | 2 | 0.5 | 65.9 |
| 1024 | 4 | 6 | 0.5 | 67.7 |
| 1024 | 4 | 4 | 0.0 | **67.8** |
| 1024 | 4 | 4 | 0.75 | 67.3 |

the same dilation factor and phase offset as the group they supplement, ensuring consistency in sampling density and spatial structure. Importantly, these points are used only as keys and values in the attention mechanism; the queries remain restricted to points within the original range interval. As a result, only the points in the interval are updated, while still attending to a complete and symmetric context. This approach maintains group separation while improving attention quality near interval boundaries.

## D  HYPERPARAMETER OPTIMIZATION

Table 6 presents a hyperparameter optimization study on the nuScenes validation set for DUA-SConv, evaluating window sizes ($K$), numbers of transformer layers ($L$), number of range intervals ($N$) in $\tilde{\mu}$, and group dropout probabilities ($P_d$) applied only during training, where $P_d = 0$ indicates no dropout. The results show that increasing $K$, $L$, and $N$ improves performance up to $K = 1024$, $L = 4$, and $N = 4$; beyond that, gains saturate and do not justify the added complexity. Dropout with $P_d = 0.5$ improves training efficiency without hurting performance, while larger values degrade accuracy.

## E  IMPLEMENTATION DETAILS

This section provides the implementation details necessary to reproduce the detection and segmentation results presented in Section 5 of the main paper.

**Detection and Segmentation Heads.**   For 3D object detection, we pair our backbone with either the CenterPoint Yin et al. (2021) or FocalFormer3D Chen et al. (2023a) detection heads. For 3D semantic segmentation, we adopt the MinkUNet Choy et al. (2019) implementation used in LinK Lu et al. (2023). In all cases, we use the official released implementations of CenterPoint, LinK, and FocalFormer3D, replacing only the backbone with our own. Detection and segmentation heads, loss weights, and post-processing procedures are left unchanged.

**Transformer Settings.**   Each LR-DUT block in the baseline model contains four Transformer layers, using 1, 1, 2, and 4 attention heads in the first through fourth layers, respectively, and a feature dropout probability of 0.1. Self-attention is computed using FlashAttention-2 Dao (2024) with a fixed window size of 1024. The feed-forward MLP in each transformer layer expands the feature dimension by a factor of four in its hidden layer, then projects it back to the original size at the output. It uses GELU activation and applies dropout with a probability of 0.1. Positional encoding is added only to the keys and queries, and is obtained by a two-layer MLP that maps each point's 3D coordinates $(x, y, z)$ to the feature dimension.

**Point Serialization.**   We adopt four space-filling curves from PointTransformer V3 Wu et al. (2024) to serialize points within each group. These patterns are listed in Table 7. Different groups are assigned different serialization patterns to increase spatial diversity in attention.

Table 7: Serialization patterns (letters correspond to Fig. 3 in PointTransformer V3).

| Label | Space-filling curve |
|-------|---------------------|
| (a)   | Transposed Hilbert  |
| (b)   | Transposed Z-order  |
| (c)   | Z-order (Morton)    |
| (d)   | Hilbert             |

**Backbone Architecture.** The full backbone consists of four DUA-SConv blocks, as illustrated in Figure 5 of the main paper. Each block downscales the spatial resolution by a factor of two in all dimensions. Every DUA-SConv block includes four transformer layers and four submanifold convolution layers with a kernel size of $3 \times 3 \times 3$, followed by batch normalization and residual connections (as in CenterPointYin et al. (2021)). In the baseline detection model, the channel dimensions are 16, 32, 64, and 128 for blocks 1-4, respectively. In the baseline segmentation model, all DUA-SConv blocks use a fixed channel dimension of 64.

**Scaled-Up Backbone.** The architecture described above corresponds to the baseline model used for both detection and segmentation. We additionally evaluate scaled-up variants. For detection, the scaled model doubles the channel dimension and the number of attention heads in each Transformer layer. For segmentation, the scaled model triples the Transformer channel dimension and number of heads, and additionally doubles the number of attention layers within each LR-DUT block.

**Training Settings.** Training hyperparameters are summarized in Table 8. All experiments use automatic mixed precision (`torch.cuda.amp`) and no gradient accumulation. For inference, we use single-model, single-scale evaluation with no test-time augmentation. All post-processing (e.g., NMS, score thresholds, voxel-to-point interpolation) exactly follows the baseline implementations.

Table 8: Optimization settings (per GPU).

| Dataset | Task | Optimizer | LR | Scheduler | Epochs | Batch |
|---------|------|-----------|-----|-----------|--------|-------|
| nuScenes | Det. | AdamW | $2 \times 10^{-4}$ | one-cycle | 20 | 16 |
| Waymo-Open | Det. | AdamW | $2 \times 10^{-4}$ | one-cycle | 36 | 16 |
| SemanticKITTI | Seg. | SGD | $1 \times 10^{-2}$ | poly ($p = 0.9$) | 80 | 4 |
| ScanNet | Seg. | SGD | $1 \times 10^{-2}$ | poly ($p = 0.9$) | 100 | 4 |
| S3DIS | Seg. | SGD | $1 \times 10^{-2}$ | poly ($p = 0.9$) | 100 | 4 |

## F  POINT DENSITY DEPENDENCE ON DISTANCE IN RGB-D DATASETS

ScanNet Dai et al. (2017) and S3DIS Armeni et al. (2016) are indoor datasets used in our evaluation (subsection 5.2), where point clouds are captured with depth (RGB-D) cameras rather than LiDAR. Despite the different sensing modalities, both exhibit the same fundamental effect relevant to our work: point density decreases approximately with the square of the range. For LiDAR, this is due to fixed angular beam spacing in spherical coordinates. For RGB-D cameras, it results from the perspective projection of a fixed-resolution image sensor, where each pixel subtends a fixed angular size in the camera's field of view. In both cases, the angular sampling causes the spacing between neighboring 3D points to grow linearly with distance, leading to denser sampling at close range and sparser sampling at longer range, which is the characteristic we model. Outdoor datasets such as nuScenes and Waymo capture objects at longer ranges than ScanNet and S3DIS. However, the relative density change for an object is comparable: the reduction in points observed when an object moves from 10 m to 40 m outdoors is similar to that when an object moves from 1m to 4m indoors, since in both cases the distance increases by a factor of four.

# G  ADDITIONAL WAYMO RESULTS: PER-CLASS AND LEVEL-1

Table 2 in the main paper reports class-averaged results on the Waymo LEVEL-2 validation set. Table 9 further presents per-class performance on LEVEL-2, while Table 10 presents LEVEL-1 results on the Waymo validation split.

Table 9: Per-Class Object Detection Results on the Waymo Validation Set (LEVEL-2)

| Methods | mAP | mAPH | Vel. | Ped. | Cyc. |
|---|---|---|---|---|---|
| AFDetV2 | 71.0 | 68.8 | 69.2 | 67.0 | 70.1 |
| SST | 67.8 | 64.6 | 65.1 | 61.7 | 66.9 |
| PV-RCNN | 66.8 | 63.3 | 68.4 | 65.8 | 68.5 |
| PV-RCNN++ | 71.7 | 69.5 | 70.2 | 68.0 | 70.2 |
| PillarNet-3 | 71.0 | 68.8 | 70.5 | 66.2 | 68.7 |
| FSD-spconv | 71.9 | 69.7 | 68.5 | 68.0 | 72.5 |
| CenterPoint | 69.8 | 67.6 | 73.4 | 65.8 | 68.5 |
| TransFusion-L | 70.5 | 67.9 | 66.8 | 66.1 | 70.9 |
| FocalFormer3D | 71.5 | 69.0 | 67.6 | 66.8 | 72.6 |
| Ours (CenterPoint) | 70.9 | 68.8 | 68.9 | 67.3 | 70.1 |
| **Ours (FocalFormer)** | 75.9 | 74.2 | 73.9 | 71.4 | 77.2 |

# H  LIMITATIONS AND FUTURE WORK

In this work, we introduce DUA-SConv as a backbone designed to combine long-range global feature extraction with fine-grained local representation learning. We evaluate the backbone within standard 3D perception settings, object detection and semantic segmentation, by integrating it with established detection and segmentation heads. However, we do not explore other long-range understanding tasks such as point-cloud classification, motion forecasting, and 3D reconstruction. Extending DUA-SConv to classification or other long-range semantic understanding tasks remains an interesting direction, and we view this as valuable future work beyond the scope of the current paper.

A further limitation of our method is that, although it reaches state-of-the-art performance in 3D detection and outdoor segmentation, it remains below the top segmentation model PTV3 Wu et al. (2024) on short-range indoor datasets. This likely stems from our range-based density equalization, which provides greater benefit in outdoor datasets where objects span a wide range of distances (near to far), resulting in large density variation. Indoor datasets have much shorter range variation, making density imbalance less pronounced.

# I  VISUALIZATIONS OF THE UDG MECHANISM

Fig. 7, Fig. 8 and Fig. 9 present qualitative visualizations of the Uniform Dilated Grouping (UDG) operation, with each figure showing a different nuScenes point-cloud example. In each figure, the full-scene point cloud is shown on the left, with two red circles highlighting similar objects at near and far distances, where the far object is roughly twice as distant as the near one.

On the upper right, we show a zoom-in of the far-range object, where object points are in blue and background points in black. Below it, we show a zoom-in of the near-range object, followed on the right by the same near object after applying UDG with a dilation factor of 4, computed using the quadratic distance-based dilation rule of UDG.

As can be seen, the near and far objects have noticeably different point densities when no dilation is applied. However, after applying the appropriate dilation factor, the near object's point cloud becomes density-aligned with the far object, resulting in a more similar geometric representation. This alignment helps the network extract features consistently across varying distances.

Additionally, we note that UDG also maintains a consistent receptive field across range. This additional benefit is not visualized here but is illustrated with real nuScenes examples in Fig. 1 of the main paper.

Table 10: Waymo validation results on LEVEL-1 difficulty (mAP / mAPH).

| Method | mAP | mAPH |
|--------|-----|------|
| DSVT | 80.3 | 78.2 |
| ScatterFormer | – | 79.7 |
| Voxel Mamba | – | 79.6 |
| SAFDNet | 81.8 | 79.9 |
| LargeKernel3D | 78.1 | 77.6 |
| Ours (F.Former) | 82.0 | 80.3 |
| Ours-Large (F.Former) | **82.8** | **81.5** |

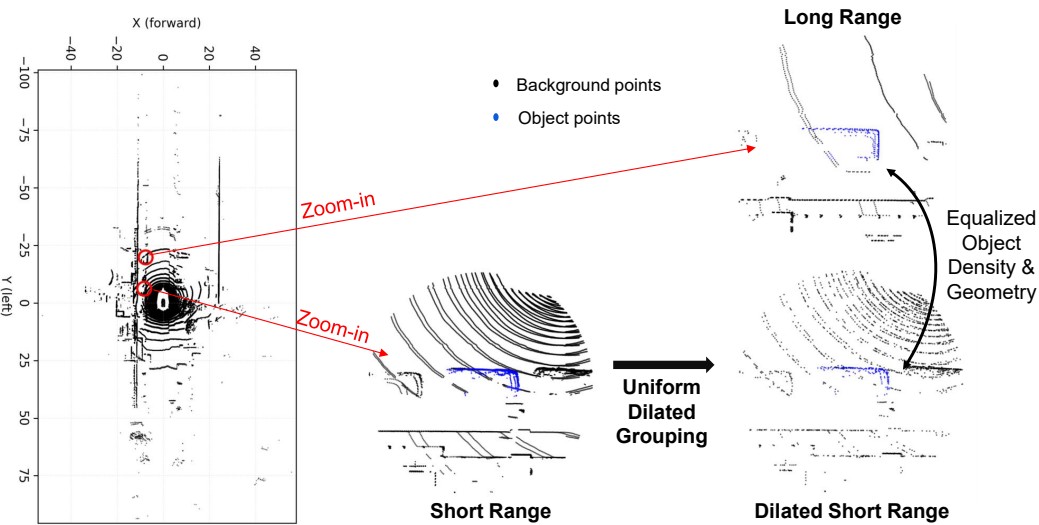

Figure 7: Qualitative visualization of the UDG operation on a nuScenes point-cloud example.

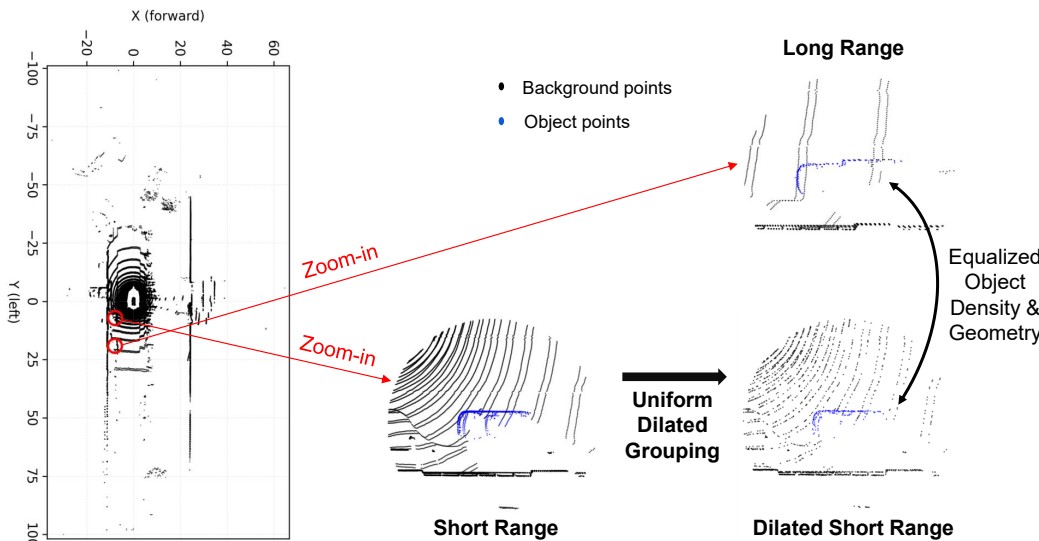

Figure 8: Qualitative visualization of the UDG operation on a nuScenes point-cloud example.

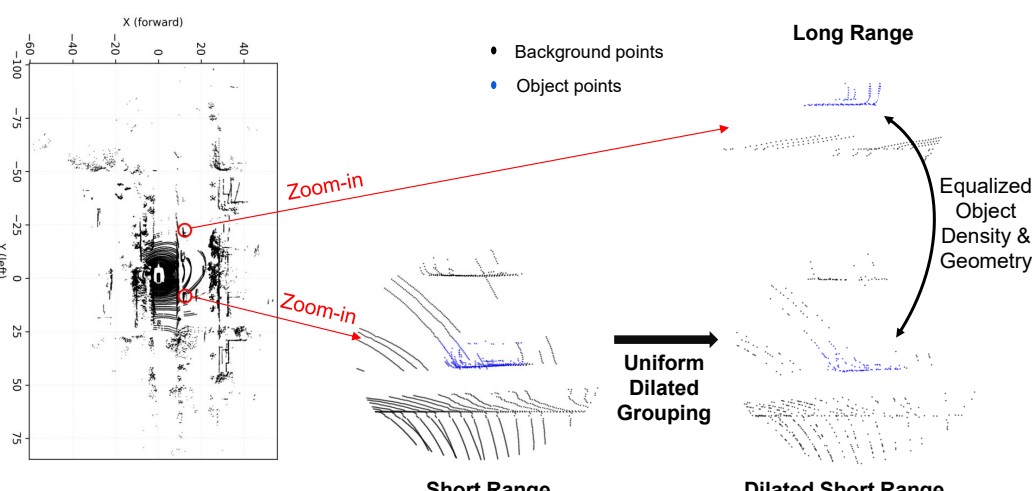

Figure 9: Qualitative visualization of the UDG operation on a nuScenes point-cloud example.

## J QUALITATIVE DETECTION EXAMPLES

Fig. 10, Fig. 11, and Fig. 12 present qualitative detection results of our DUA-SConv backbone with the FocalFormer detection head, shown alongside the ground-truth bounding boxes. These examples demonstrate the close alignment between our detections and the ground truth, consistent with the quantitative performance reported in Section 5.1 of the main paper.

**Ground Truth**

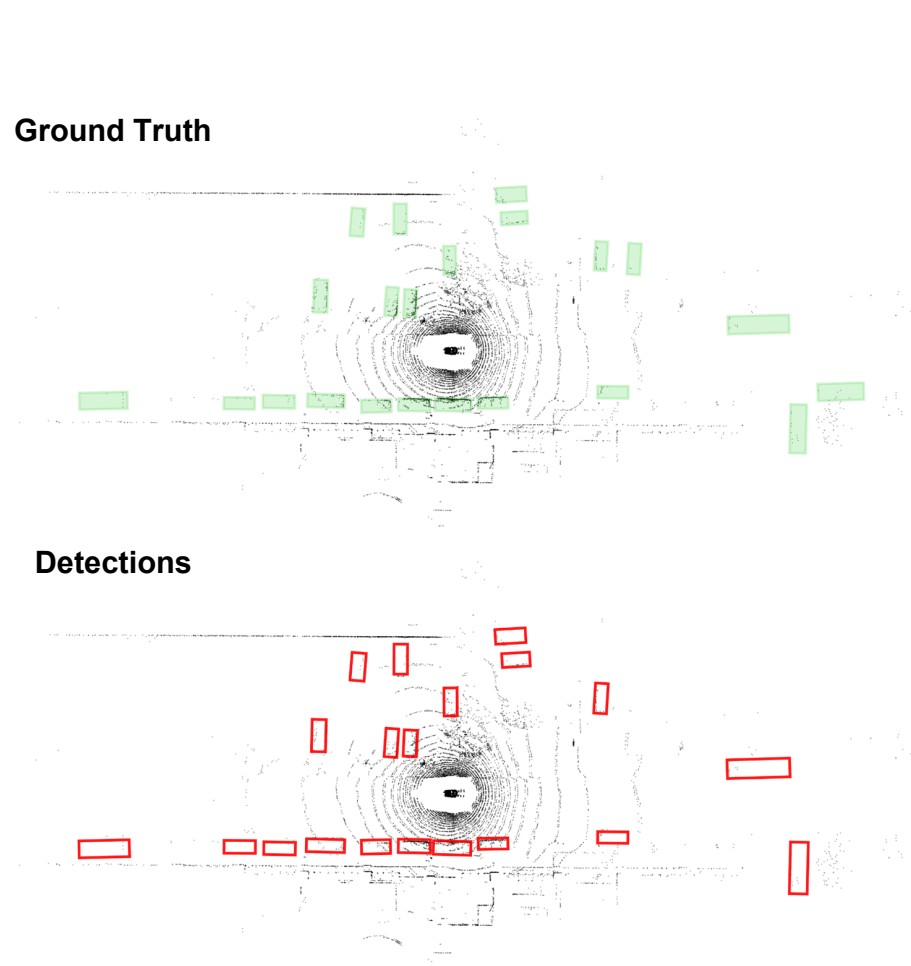

**Detections**

Figure 10: Qualitative detection results on nuScenes point clouds. The upper part shows ground-truth bounding boxes, and the lower part shows our detections produced using the DUA-SConv backbone with the FocalFormer detection head.

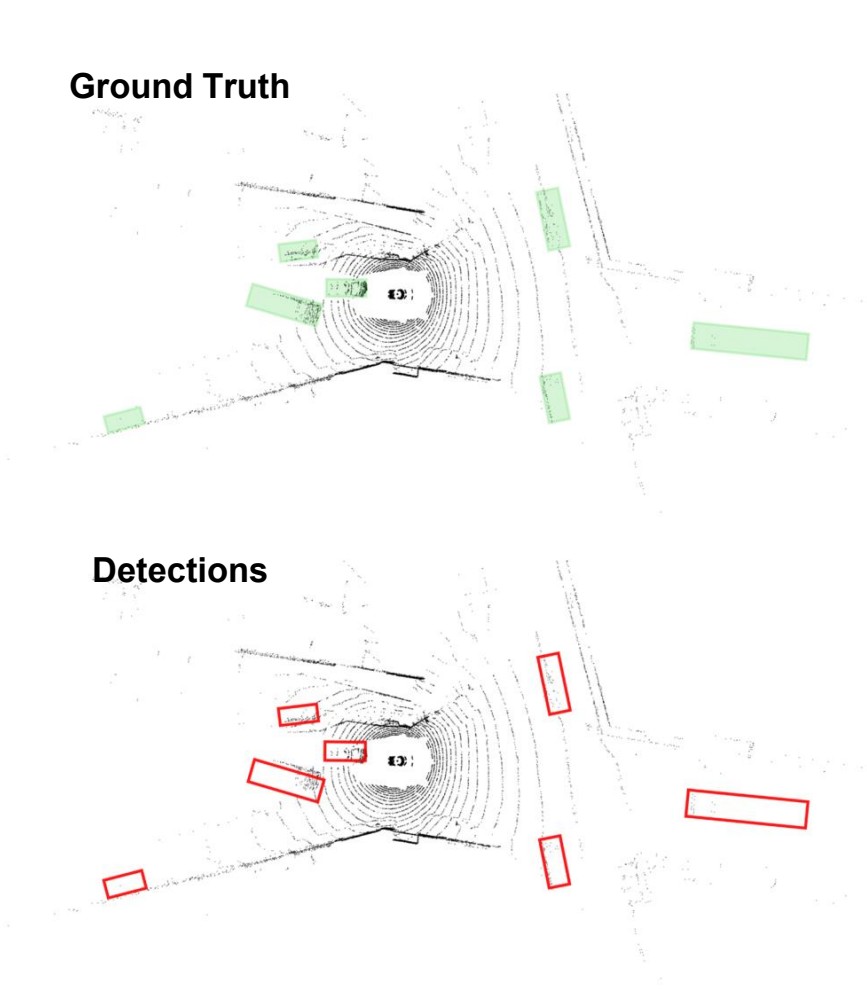

Figure 11: Qualitative detection results on nuScenes point clouds. The upper part shows ground-truth bounding boxes, and the lower part shows our detections produced using the DUA-SConv backbone with the FocalFormer detection head.

**Ground Truth**

**Detections**

Figure 12: Qualitative detection results on nuScenes point clouds. The upper part shows ground-truth bounding boxes, and the lower part shows our detections produced using the DUA-SConv backbone with the FocalFormer detection head.

