# OpenReview forum: "An Efficient Global-Local Feature Extraction Architecture for 3D Point Clouds"
_ICLR.cc/2026/Conference — Submitted to ICLR 2026_

### Official Review · Reviewer_RTGD · 2025-10-22

**Soundness:** 3
**Presentation:** 3
**Contribution:** 3
**Rating:** 4
**Confidence:** 4

**Summary:**

DUA-SConv is a hybrid 3D backbone that combines attention and sparse convolution to balance global context and local detail in LiDAR-based 3D detection and segmentation. It introduces Dilated Uniform Attention to capture wide-range contextual information efficiently, followed by sparse convolution for precise local feature recovery. By stacking these lightweight DUA-SConv blocks, the model achieves high accuracy with reduced computational cost and parameter count.

**Strengths:**

**[S1] Clear motivation and supporting evidence. **

The paper provides a clear motivation by emphasizing the importance of modeling long-range receptive fields in 3D point processing. Their discussion on the limitations of conventional convolutions effectively supports the need for their proposed approach. The theoretical justification is coherent, linking the receptive field expansion to improved context aggregation. Empirical results further reinforce their argument, showing noticeable improvements in capturing global scene structures. Overall, their claim regarding the long-range receptive field is convincing and well-supported by both analysis and experiments.

**[S2] Effective performance on nuScene dataset.**

The proposed model demonstrates strong performance on the **nuScenes** dataset, effectively handling large-scale outdoor environments. Its ability to process sparse and wide-range point distributions highlights the robustness of the convolution design. Quantitative results show clear improvements over previous baselines, indicating consistent advantages in outdoor scenarios. The authors also emphasize that their method maintains efficiency while preserving accuracy across long-range interactions. Together, these results confirm that the model is particularly well-suited for outdoor perception tasks where spatial coverage is critical.

**Weaknesses:**

**[W1] Missing PTv3 and Sonata baselines on nuScenes**

The authors claim that their proposed convolution demonstrates strong capability in modeling long-range interactions. While the presented results are promising, this claim would be more convincing if the authors included comparisons against **PTv3** and **Sonata** on outdoor datasets such as **nuScenes**. Additionally, **Sonata** is missing from the evaluations on **S3DIS** and **ScanNet**, despite being published at **CVPR 2025**. Including it would provide a fairer and more comprehensive comparison.

**[W2] Missing qualitative results**

Although the proposed method achieves competitive quantitative performance, the paper lacks **qualitative results** or detailed visual analysis. The authors should include qualitative comparisons with prior methods on datasets such as **nuScenes** and **ScanNet**, which would help illustrate the qualitative advantages and better support their quantitative claims.

**[W3] Slightly worse performance on SemSeg datasets**

The proposed method underperforms **PTv3** on indoor datasets, indicating potential limitations in handling short-range geometric structures. This suggests that the model may not generalize well to indoor or densely cluttered environments. To provide a more comprehensive understanding, the authors could also include experiments on **classification** or other **long-range understanding** tasks to validate the effectiveness of their proposed convolution. Moreover,

**Questions:**

- In **Figure 4**, the groups appear duplicated when N=64. Is there a specific reason why **azimuth** and **elevation** are partitioned with overlaps rather than being uniquely divided? Also, could the authors clarify what the **index values** in the figure represent?
- How were **azimuth** and **elevation** values computed for **ScanNet**? Depending on the normalization procedure of point clouds, these values may vary significantly. Please clarify the computation method and normalization scheme used.

---

> ### Author Response · Authors · 2025-11-23
>
> Thank you for your time and helpful feedback.
>
> **Response to comment on missing PTv3 and Sonata baselines.**
>
> PTv3 is already included in our comparisons in Table 3 of the paper. While it achieves higher segmentation accuracy, it does so with a model more than four times larger than ours, placing it in a very different model-capacity regime.
>
> Regarding Sonata, it is a self-supervised pretraining framework rather than a standalone backbone. Sonata is designed to improve representations learned by point-cloud encoders and can be applied to many backbone families. It is therefore orthogonal to our contribution and could, in principle, be applied on top of our model as well.
>
> In this work, we aim to isolate and evaluate the contribution of our backbone (DUA-SConv) without additional gains from self-supervised learning. In the final paper, we will expand the related-work section to include SSL methods, particularly Sonata, and highlight their potential to further enhance point-cloud models, including ours.
>
>
> **Response to comment on missing qualitative results.**.
>
> We have added qualitative visualizations to complement the quantitative results. Appendix H now includes visualizations of the Dilated Uniform Attention (DUA) operation on real point clouds, illustrating the effect of the proposed grouping and attention mechanism. Appendix I provides qualitative examples of object detection using our DUA-SConv backbone. These visualizations highlight the qualitative behavior of our method and further support the quantitative results reported in the main paper.
>
> **Including experiments on classification or other long-range understanding tasks.**
> We note that the baseline reference methods included in our comparisons focus on object detection and/or semantic segmentation, and do not evaluate point-cloud classification, which is a distinct task. Our work is likewise designed as a backbone for object detection and segmentation, consistent with these standard evaluation settings. Extending DUA-SConv to point-cloud classification or other long-range understanding tasks is indeed an interesting direction, and we view this as valuable future work beyond the scope of the current paper.
>
> **Response to the question on the duplicated groups in Figure 4 when N=64 and clarification of the index values.**
>
> Each index value in Figure 4 corresponds to the identifier of a dilated group. All azimuth–elevation cells assigned to the same group share the same color and the same index, because they represent the points that will be processed together by a single Transformer window. Hence the apparent duplication when N=4,16,64 is expected. The figure shows only a cropped portion of the full azimuth-elevation range for visualization purposes. The grouping pattern repeats periodically as the azimuth and elevation span increases, so identical index values reappear across the extended field of view. We will clarify the meaning of these index values and the repeated pattern in the final paper version.
>
> **Response to the question regarding azimuth and elevation computation in ScanNet.**
>
> For ScanNet, azimuth and elevation are computed by converting each point to spherical coordinates using the point-cloud centroid as the reference origin.

---

> ### Comment · Reviewer_RTGD · 2025-11-26
>
> [R1]
> To clarify, regarding [W1], I did not claim that PTv3 is missing from Table 3. My primary concern is that PTv3 results on the nuScenes object detection (Table 1) is missing, despite the authors including other baselines such as SphereFormer, which is known to perform reliably across various 3D benchmarks. Simply comparing on segmentation task is not enough to claim their superiority since PTv3 outperforms F.Former on nuScene (semantic segmentation). I apologize if my review regarding [W1] was unclear.
>
> Also, if the authors argue that the performance gap is due to model size, they should demonstrate that a scaled-up version of their model—closer to the size of PTv3—achieves comparable performance. If this is not feasible, the authors should explicitly acknowledge that their method may underperform in short-range scenarios.
>
> I also agree that Sonata is out of scope, given its self-supervised training paradigm; I apologize for the earlier misunderstanding regarding its omission.
> Additionally, although the authors stated that the Related Work section was updated, the revised manuscript does not reflect these changes. This section must be updated in the revised version itself, as reviewers base their final evaluations on the rebuttal version, not the camera-ready version.
>
> [R2]
> Thank you for updating the qualitative results. My concern for [W2] is fully resolved.
>
> [R3]
> I accept the authors’ explanation. However, I strongly encourage adding this discussion to a Limitations and Future Work section.
>
> [R4, R5]
> Understood. The design choices now seem reasonable.
>
> As a minor suggestion, I could not easily identify which parts were revised. I recommend marking revisions using a different color so reviewers can quickly verify the updates.
> Additionally, I noticed an overlapping citation on page 14; this should be corrected.

---

> > ### Author Response · Authors · 2025-11-30
> >
> > **1. Response on missing PTv3 results on nuScenes object detection (Table 1).**
> >
> > We thank the reviewer for the clarification. We have now added PTv3 results for nuScenes
> > object detection to Table 2 in the revised manuscript. The comparison shows that our base
> > model, roughly 4× smaller than PTv3, as well as several other state-of-the-art detectors,
> > outperforms PTv3 on object detection, while PTv3 achieves strong performance in
> > segmentation (Table 3). This highlights the challenge of designing a single backbone
> > architecture that performs well across tasks. Our method offers an efficient backbone that
> > achieves strong and competitive performance on both detection and segmentation.
> >
> > **2. Response on scaled up version of our model for segmentation task:**
> >
> > We thank the reviewer for this suggestion. We have evaluated a scaled-up version of our
> > backbone, referred to as Ours-H, for semantic segmentation. This model triples the
> > Transformer feature dimension and number of attention heads, and doubles the number of
> > attention layers, increasing the parameter count of the baseline model from 10.8M to
> > 44.7M - comparable to PTv3 (46.2M). The results are presented in Table 3 of the revised
> > manuscript.
> >
> > Our scaled up model (Ours-H) outperforms PTV3 and other reference methods on long-range outdoor datasets (nuScenes, KITTI, Waymo) and approaches, though still below, PTV3
> > performance on short-range indoor datasets (ScanNet, S3DIS). This likely stems from our
> > range-based density equalization, which provides greater benefit in outdoor datasets where
> > objects span a wide range of distances (near to far), resulting in large density variation.
> > Indoor datasets have much shorter range variation, making density imbalance less
> > pronounced.
> >
> > **3. Response to comment on updating the related work:**
> >
> > We have now updated the Related Work section directly in the revised manuscript,
> > including the addition of relevant self-supervised learning approaches such as Sonata. We
> > apologize for the earlier misunderstanding, our previous response referred to camera-ready
> > changes, but we have now incorporated these updates into the current revision that the
> > reviewers evaluate.
> >
> > **4. Response on adding Limitations and Future Work discussion.**
> >
> > Following the reviewer’s recommendation, we have added a Limitations and Future Work
> > section in Appendix H of the revised manuscript. This section discusses potential extensions
> > of our backbone to additional long-range understanding tasks (e.g., classification) and
> > explicitly acknowledges the reviewer’s earlier point that, while our method outperforms
> > PTv3 in outdoor (long-range) detection and segmentation, it still falls short of PTv3 on
> > indoor (short-range) segmentation benchmarks. This likely stems from the fact that our range-normalized design benefits most from scenarios where point-density changes significantly with distance, an effect that is much more pronounced in outdoor LiDAR settings compared to indoor scans.
> >
> > **5. Response on highlighting revisions and fixing the citation issue.**
> >
> > We have marked the updates in the revised manuscript in blue for easy identification, and
> > we have verified that there is no longer an overlapping citation.

---

### Official Review · Reviewer_AzB2 · 2025-10-28

**Soundness:** 3
**Presentation:** 3
**Contribution:** 2
**Rating:** 4
**Confidence:** 5

**Summary:**

The paper presents an efficient transformer design that unifies global-local LiDAR perception through uniform grouping, localized attention, and implicit relative positional encoding—achieving strong accuracy with scalable computation.

**Strengths:**

1. The paper is well-motivated with a clear explanation of the underlying challenge in LiDAR perception. The authors use visualizations effectively to illustrate the density imbalance across different ranges and to show why uniform grouping and local-global modeling are necessary.
2. The method achieves state-of-the-art results on both NuScenes and Waymo Open Dataset

**Weaknesses:**

1.Although the paper lists three main contributions, most of them boil down to the introduction of the DUA (Dilated Uniform Attention) module. The overall methodological novelty feels incremental, as the proposed framework mainly adapts existing attention mechanisms to LiDAR range representations rather than introducing a fundamentally new idea.

2.The DUA module itself is not highly innovative, it essentially performs standard attention operations on the range image domain, similar to what has been explored in prior transformer-based LiDAR perception works. The conceptual leap from previous designs is therefore relatively small.

3.The model requires transformations between the range image and sparse point cloud spaces, along with additional grouping operations. These steps likely introduce notable latency. The reported 117 ms inference time is considerably slower than recent efficient LiDAR transformers such as HEDNet and ScatterFormer, while the accuracy improvement is relatively modest, raising concerns about the overall efficiency–performance trade-off.

He, C., Li, R., Zhang, G., & Zhang, L. (2024). ScatterFormer: Efficient Voxel Transformer with Scattered Linear Attention. In Proceedings of ECCV 2024

**Questions:**

1. Since the model focuses on efficient global-local feature aggregation, it would be interesting to see whether the approach generalizes beyond detection to dense prediction tasks such as semantic or instance segmentation.
2. I’m curious how much latency is introduced when transforming the point cloud into the range-view representation and performing the grouping operations, before converting it back to the sparse point cloud space.

---

> ### Author Response · Authors · 2025-11-23
>
> Thank you for your time and helpful feedback.
>
> **On the novelty of the method beyond adapting existing attention mechanisms to LiDAR range representations.**
>
> While the Dilated Uniform Attention (DUA) module is a key component of our method, its contribution is not a simple adaptation of existing attention mechanisms to LiDAR range views. DUA introduces a novel range-aware dilated grouping strategy that forms uniform-density point groups directly in Cartesian voxel space, rather than in spherical or range-image coordinates. This ensures that attention operates with consistent geometric properties and receptive fields across distance, addressing a core LiDAR bottleneck that prior range-view or spherical representations do not resolve, as object shape distorts significantly with distance in those spaces.
>
> Furthermore, the integration of DUA with sparse convolution in the DUA-SConv block is an essential architectural contribution. DUA provides coarse, range-invariant global context, and the subsequent sparse convolution injects this context back into the full-resolution 3D structure, enabling the recovery of fine geometric detail. This two-stage design is fundamentally different from simply adapting attention to LiDAR: it provides efficient long-range context modeling while preserving high-resolution spatial fidelity.
>
>
> **Response to the comment that DUA performs standard range-image attention.**
> We would like to clarify that DUA does not operate in the range-image domain. Our pipeline voxelizes the point cloud in Cartesian space, and both attention and convolution are performed entirely in this sparse-tensor domain. The only use of spherical coordinates is to compute a range-dependent dilation factor for grouping, which equalizes point density across distances.
>
> This design is fundamentally different from range-image attention approaches, where angular sampling is uniform but object geometry is highly distorted with distance, nearby objects span many pixels, while distant objects collapse into very few. Such distortions produce range-dependent receptive fields and unstable feature representations. In contrast, our dilated grouping produces groups with approximately uniform density and consistent geometric structure across ranges. This yields attention windows with fixed receptive fields and preserves object shape in Cartesian space, avoiding the geometric inconsistencies inherent in range-image or spherical representations.
>
> **Response to comment on computation overhead of transformations between the range image and sparse point cloud spaces, grouping operations, and the overall efficiency-performance trade-off.**
>
> We would like to clarify that DUA-SConv operates entirely in the voxelized sparse-tensor domain and does not alternate between range-image and sparse representations. The structured dilation uses each point’s spherical coordinates only to assign it to a dilated group, but all attention and convolution operations are performed in sparse tensor Cartesian space. This grouping step does not contribute significantly to the overall latency; the dominant runtime comes from the window attention and sparse-convolution layers.
> For the efficiency-performance trade-off analysis please refer to our response to Reviewer nrxi under “Comparison of performance and efficiency for our baseline and scaled-up models against recent state-of-the-art methods”. This section includes comparison table with HEDNet, ScatterFormer, and other recent state-of-the-art methods. HEDNet and ScatterFormer are indeed highly efficient, but several CVPR 2025 methods, such as FSHNet and UniMamba, achieve incremental accuracy gains at the cost of increased runtime. Our method follows this same trend: achieving accuracy improvements over efficient baselines requires additional compute once the performance frontier is already high.
>
> **Response to comment on generalizes beyond detection to dense prediction tasks such as semantic or instance segmentation.**
>
> We would like to clarify that our paper already includes semantic segmentation experiments across multiple datasets.
> Please refer to Section 5.2 and Table 3, which show that the proposed backbone generalizes effectively beyond object detection to dense prediction tasks.
>
> **On latency introduced by grouping and reverting to full resolution.**
> The table below reports average profiling results for the main components of the DUA module. The grouping step, which assigns points to dilated groups based on their spherical coordinates, takes only 0.7 ms (0.6% of runtime). The subsequent group-combining step adds 4.1 ms (3.5%). Overall, these operations contribute only a small latency overhead relative to the full DUA-SConv block.
>
> |Operation|Runtime [ms]|% of DUA-SConv|
> |---|---|---|
> |Partitioning point cloud to groups (Grouping)|0.7|0.6%|
> |1D Serialization|1.4|1.2%|
> |Window Attention (FlashAttention)|34.6|29.6%|
> |Groups combining (including inverse mapping)|4.1|3.5%|

---

> > ### Comment · Reviewer_AzB2 · 2025-11-27
> >
> > I appreciate that the authors have addressed my concern about grouping voxel indices using spherical coordinates. However, while the proposed method yields moderate gains (+1.3 points on Waymo L2 and +0.4 NDS on nuScenes), these improvements come at the cost of a substantial increase in inference latency (2~3× slower than HEDNet and ScatterFormer). Given the limited performance uplift relative to the significant speed penalty, I prefer to retain my original score.

---

> > > ### Author Response · Authors · 2025-11-30
> > >
> > > **Response on significance of the performance gain.**
> > >
> > > We would like to note that both Waymo and nuScenes are highly saturated benchmarks, where
> > > year-to-year progress at the top is typically below one point. Recent works published in top-tier
> > > conferences, including LION, UniMamba, Voxel Mamba, FSHNet and SAFDNet, also report
> > > incremental improvements, often smaller than ours, while introducing runtime overheads that
> > > are 2-3× larger than HEDNet and ScatterFormer. In this context, our +1.3 mAPH gain in Waymo
> > > L2 is competitive and consistent with current state-of-the-art progress in 3D perception.
> > >
> > > Additionally, increasing Waymo L2 mAPH from 73.8 (ScatterFormer ) to 75.1 (ours) reduces the
> > > remaining error from 26.2% to 24.9%, which is a ~5% reduction in detection failures. At
> > > deployment this typically translates to hundreds of thousands of additional correctly detected
> > > objects per million frames, reflecting a meaningful improvement for safety-critical perception.
> > >
> > > Beyond the absolute performance values, we hope the reviewer also recognizes the conceptual
> > > contribution of Uniform Dilated Grouping (UDG), which introduces a new perspective on range-normalized geometry in point clouds, as well as DUA-SConv as a full architectural realization of
> > > this idea within a complementary Transformer-convolution framework. We believe this
> > > contribution benefits the community not only through the performance demonstrated here,
> > > but also as a foundation that future work can extend, optimize, and integrate into even more
> > > efficient architectures.

---

### Official Review · Reviewer_mcbN · 2025-10-29

**Soundness:** 3
**Presentation:** 3
**Contribution:** 3
**Rating:** 6
**Confidence:** 3

**Summary:**

This paper proposes DUA-SConv, an efficient hybrid backbone for 3D point cloud processing that addresses the challenge of capturing both global context and local detail. The core contribution is a novel "Uniform Dilated Grouping" (UDG) mechanism that applies range-dependent dilation to compensate for the non-uniform density of LiDAR data. This allows an efficient serialized transformer to learn coarse global context from large, consistent spatial regions. This global context is then refined by a 3D sparse convolution to capture fine-grained local features. Experiments demonstrate its effectiveness.

**Strengths:**

1.	The Uniform Dilated Grouping (UDG) mechanism is a novel and technically sound method for "equalizing" point cloud density before applying attention. This directly addresses a key limitation of prior window-based transformers, whose receptive fields are spatially inconsistent (Fig. 1b). The complementary design, using attention for coarse-global context and sparse convolution for fine-local refinement, is well-motivated and elegant.
2.	Extensive experimentation and ablation studies validate the effectiveness of the proposed method.
3.	This paper is written and  organized well.

**Weaknesses:**

1.	The paper lacks intuitive visualizations of the UDG mechanism in action. While Figure 4 shows the indices of the groups, it does not provide a qualitative visualization of what a "dilated group" of points actually looks like in a real point cloud, especially when contrasted with a "naive" group. Adding such a visualization would significantly help readers understand the practical effect of UDG.
2.	The key components of the LR-DUT module, such as point serialization and the K/Q-only positional encoding, appear heavily adopted from Point Transformer V3.  The core innovation is clearly the Uniform Dilated Grouping (UDG). It would be more precise to frame the main contribution as the novel integration of UDG with an existing serialized attention framework, rather than implying the entire LR-DUT block is a novel invention.
3.	It will be more convincing to add more advanced method UniMamba[1] in Tab.1.

[1] UniMamba: Unified Spatial-Channel Representation Learning with Group-Efficient Mamba for LiDAR-based 3D Object Detection. CVPR 2025

**Questions:**

Refer to the Weakness.

---

> ### Author Response · Authors · 2025-11-23
>
> Thank you for your time and helpful feedback.
>
> **Intuitive visualizations of the UDG mechanism:**
> Following the reviewer’s suggestion, we have added qualitative visualizations illustrating what a ‘dilated group’ looks like in a real point cloud. These visualizations help clarify the practical effect of UDG. Please refer to Appendix H in the supplementary material.
>
> **Response to: "The key components of the LR-DUT module, such as point serialization and the K/Q-only positional encoding, appear heavily adopted from Point Transformer V3. The core innovation is clearly the Uniform Dilated Grouping (UDG). It would be more precise to frame the main contribution as the novel integration of UDG with an existing serialized attention framework, rather than implying the entire LR-DUT block is a novel invention."**:
>
> While our LR-DUT module adopts the serialization strategy introduced in Point Transformer V3 (PTv3), both its positional encoding mechanism and its role within the overall backbone differ substantially. PTv3 uses conditional positional encoding (xCPE),  whereas LR-DUT employs efficient relative positional encoding by adding absolute positional encodings only to the Keys and Queries (not the Values), as detailed in Appendix B. Furthermore, the proposed DUA-SConv block integrates LR-DUT for coarse global feature extraction followed by full-resolution sparse convolution for local feature refinement, which is a processing flow that differs from PTv3.
> Following the reviewer’s comment, we will clarify these distinctions between our method and PTv3 in the final version of the paper, and make explicit that the novelty of the LR-DUT component lies in integrating UDG with a serialized attention framework, rather than implying that the entire LR-DUT module is newly invented.
>
> **Response to reviewer comment:** “It will be more convincing to add more advanced method UniMamba[1] in Tab.1.”:
>
> Please refer to our response to Reviewer nrxi under “Comparison of performance and efficiency for our baseline and scaled-up models against recent state-of-the-art methods”. This section includes a direct comparison with several recently published SOTA methods, including UniMamba.

---

### Official Review · Reviewer_nrxi · 2025-10-31

**Soundness:** 3
**Presentation:** 3
**Contribution:** 2
**Rating:** 4
**Confidence:** 3

**Summary:**

The paper presents an architecture that integrates the local feature extraction capability of 3D sparse convolutions with the long-range contextual modeling of dilated attention. This combination enables more effective feature learning from point clouds, which often suffer from uneven density compared to other data modalities. The method builds upon established techniques (3D sparse convolutions, point cloud serialization, transformers, and positional encoding) and introduces Uniform Dilated Grouping (UDG) strategy which forms the foundation of the proposed DUA-SConv module, the core component of the architecture. Experimental results on popular benchmarks demonstrate that the proposed network outperforms other models of similar size and achieves competitive performance compared to larger architectures.

**Strengths:**

The paper is overall clear, well-structured, and easy to follow.

The motivation for integrating modules that capture local geometric details (via 3D sparse convolutions) with those that model long-range context (via dilated transformers) is well-justified.

The proposed UDG effectively partitions point clouds into groups of approximately uniform spatial size and density which is an interesting and practical solution to the varying point density problem of LiDAR data. The authors combine this component with well-established and effective techniques in a structured and coherent manner to construct the overall architecture.

Results on many popular benchmarks are reported.

**Weaknesses:**

The proposed method has some similarity to the Neurips 2024 paper "LION: Linear Group RNN for 3D Object Detection in Point Clouds", which applies linear RNN operators on grouped features within a window-based framework. The current paper seems to extend this learning paradigm to window transformers and the 3D Sparse Convolution is very similar to the 3D sub-manifold convolution of LION in capturing local information. This undermines the novelty, especially as performance improvements are also limited. Can the authors provide a more thorough comparison with LION?

Since the paper primarily emphasizes efficiency, it would benefit from additional comparisons of FLOPs, memory consumption, and runtime against Transformer-based models (e.g., PTv3, SphereFormer), Mamba-based models (e.g., Voxel Mamba) and LION to better justify the efficiency–performance trade-offs. The authors should include FLOPs, memory, and runtime comparisons, along with results from scaled-up versions of the proposed model. This will further strengthen the paper.

Although the proposed architecture has the advantage of a smaller model size (i.e., fewer parameters), it does not appear to achieve state-of-the-art performance. While achieving SOTA results is not strictly necessary, an analysis of scaling strategies and an evaluation of a larger version of the model compared to current SOTA methods would provide valuable insight into the design’s potential capabilities.

The Waymo Level 1 results are not reported. Can authors include this and keep result reporting consistent with Voxel Mamba, SAFDNet, LION as well as the UniMamba & FSHNet papers cited below.

Comparisons do not include the most recent methods. The following papers perform better than the current method. Can you provide comparisons to these baselines:

[1] S Liu et al. "FSHNet: Fully Sparse Hybrid Network for 3D Object Detection." CVPR 2025.

[2] X Jin et al. "UniMamba: Unified Spatial-Channel Representation Learning with Group-Efficient Mamba for LiDAR-based 3D Object Detection." CVPR 2025.

[3] Z Liu et al. "LION: Linear Group RNN for 3D Object Detection in Point Clouds." NeurIPS 2024.

Why does adding positional encoding to K, Q, and V lead to lower performance compared to adding it only to K and Q (as shown in Table 4)?

There are a few minor writing issues, though they do not affect the overall understanding of the paper. In particular, the terms “dilation” and “dilution” should be used consistently (e.g., the caption of Figure 3 should use “dilation factor” to match the terminology in the main text). Additionally, the column names in Table 5 are missing and should be included for clarity.

**Questions:**

Please see the Weaknesses section. I am open to changing my rating if the authors can address my comments.

---

> ### Author Response · Authors · 2025-11-23
>
> Thank you for your time and helpful feedback.
>
> **Comparison between our method and LION.**
>
> While both our method and LION combine a global operator with local 3D convolutions, the methods through which they achieve this, and their technical contributions, differ substantially.
> 1. **Different long-range feature extraction method.**
> LION replaces attention with a linear RNN to overcome the limited receptive field of standard Transformers caused by their quadratic window cost. In contrast, we retain the Transformer and resolve this limitation with Uniform Dilated Grouping, enabling small-window attention to capture long-range context while preserving the advantages of parallel pairwise interaction and permutation-invariant geometric representation over order-dependent RNNs.
>
> 2. **Range-invariant receptive fields and geometric features.**
> LION groups points purely by spatial proximity, so sequence density varies dramatically with range. Our UDG explicitly equalizes point density, ensuring that the Transformer always processes groups with consistent geometric properties and a range-invariant receptive field.
>
> 3. **Different purpose of sparse convolutions.**
> In LION, sparse convolutions mainly restore spatial information lost from RNN ordered serialization. In our method, they instead complement UDG by refining local geometry and spreading global context.
>
> 4. **Different scope and applicability.**
>  LION is detection-oriented and optimized for object-level processing. Our backbone is designed for both detection and semantic segmentation, which require finer geometric detail at the voxel level. This broader capability reflects a distinct design space.
>
> **Comparison of performance and efficiency for our baseline and scaled-up models against recent state-of-the-art methods (UniMamba, FSHNet, LION), including Waymo L1 results.**
>
> The table below compares Transformer (Tran.), SSM, and sparse-convolution backbones, including UniMamba, FSHNet, LION, Voxel Mamba, as well as our base model (“Ours”) and scaled-up variant (“Ours-L”). The scaled-up model doubles the Transformer feature dimension and attention heads, increasing parameters from 10.8M to 16.2M - bringing its size in line with leading methods.
> Our baseline already provides strong accuracy with low compute (e.g., higher accuracy than LION and Voxel Mamba with lower runtime). Our scaled model further improves accuracy, performing mostly superior to recent SOTA methods (UniMamba, FSHNet) while adding only moderate increases in parameters and latency.
> It is important to note that FSHNet, UniMamba, and LION are designed specifically for 3D object detection, whereas our backbone is applicable to both detection and semantic segmentation. We demonstrate strong performance in both tasks, including segmentation benchmarks that require fine-grained spatial detail and long-range context. Notably, methods such as Point Transformer V3 achieve strong segmentation accuracy but underperform on detection (see table below), underscoring the challenge of designing a single architecture that performs well across tasks.
>
> |Method|Type|Waymo-L1-Val|Waymo-L2-Val|nuScenes-Val|nuScenes-Test|#Param (M)|Latency (ms)|
> |---|---|:--:|:--:|:--:|:--:|---:|---:|
> | | |**mAP/mAPH**|**mAP/mAPH**|**mAP/NDS**|**mAP/NDS**| | |
> |Ours (F.Former)|Tran.|82.0/80.3|75.9/74.2|67.8/72.0|70.1/73.3|10.8|119|
> |Ours-L (F.Former)|Tran.|82.8/81.5|77.0/75.1|68.9/72.8||16.2|137|
> |SphereFormer|Tran.||||65.5/70.7|32.3||
> |FSHNet_bs|Tran.||77.1/74.9|68.1/71.7||13.1|123|
> |DSVT|Tran.|80.3/78.2|74.0/72.1|66.4/71.1|68.4/72.7|7.1|100|
> |ScatterFormer|Tran.|--/79.7|--/73.8|68.3/72.4||12.6|45|
> |PTV3|Tran.||--/73.0|||40||
> |LION|SSM||75.1/73.2|68.0/72.1|69.8/73.9|16.1|152|
> |UniMamba|SSM||76.13/74.11|68.5/72.6|70.2/74.0|16.3|121|
> |Voxel Mamba|SSM|--/79.6|--/73.6|67.5/71.9|69.0/73.0|15.1|182|
> |SAFDNet|Conv.|81.8/79.9|75.7/73.9|66.3/71.0|68.3/72.3|15.8|106|
> |LinK|Conv.|||63.6/69.5|66.3/71.0|10.3|109|
> |LargeKernel3D|Conv.|78.07/77.61|69.81/69.38|63.3/69.1|65.3/70.5|18|83|
> |HEDNet|Conv.||75.3/73.4|66.7/71.4|67.7/72.0|15.3|67|
>
> **Why adding positional encoding to K, Q, and V performs worse than only to K and Q:**
>
> In attention, where the output is a weighted sum of neighboring Value vectors, we want the relative position between the query point and each neighbor to affect the weights, rather than the values being averaged. Since attention weights come from the dot product between K and Q, adding absolute positional encoding to them allows the weights to capture relative spatial relationships directly through that dot product, without extra cost. Keeping V position-free preserves spatial invariance, whereas adding position to V injects absolute coordinates into the weighted sum, causing positional signals to overwhelm feature content and lowering performance (Table 4 in the paper). For further details, please refer to Appendix B in the supplementary material.
>
> The minor writing issues have been corrected in the revised paper.

---

> > ### Comment · Reviewer_nrxi · 2025-11-26
> > **Comments on response**
> >
> > I appreciate the authors' efforts for providing a detailed response and performing the additional results. My comments have been mostly addressed apart from providing FLOP comparison. Are you able to compare FLOPs to at least the nearest competitors?
> >
> > I will raise my score to 6.

---

> > > ### Author Response · Authors · 2025-11-30
> > >
> > > Thank you for the positive feedback and for raising your score.
> > > We plan to include FLOPs comparisons in the camera-ready version.

---

### Author Response · Authors · 2025-11-30

Dear Area Chair,
We appreciate your time and role in evaluating our submission. In light of the updated decision
procedure, we are providing a focused summary of the updates made to the paper. We have
thoroughly addressed all reviewer comments and revised the paper accordingly. A detailed
rebuttal has been submitted, and a revised manuscript has been uploaded with the updates
clearly marked in blue. The main improvements are summarized below:

**1. Scaled-up backbone evaluation - key new results**

The original submission demonstrated strong performance with a compact model, and
reviewers nrxi and RTGD requested evaluation at a larger capacity similar to top-performing
methods. In response, we trained and evaluated scaled-up versions of our backbone for both
detection and segmentation, with parameter sizes comparable to leading approaches (Tables 1,
2, 3, 5). These larger models substantially improve over the baseline, achieving state-of-the-art
object detection across most metrics, state-of-the-art outdoor segmentation, and competitive indoor
segmentation, where we are only slightly below SOTA PTV3 [5] in indoor segmentation.

Importantly, while many methods excel at either detection or segmentation, our backbone
performs strongly across both tasks, highlighting its versatility. For example, Point Transformer
V3 (PTV3) achieves strong segmentation performance (Table 3) but falls behind leading
methods in object detection (Table 2), whereas ours remains consistently strong across both.

**2. Stronger Detection Benchmark with New References**

Per reviewers nrxi and mcbN suggestions, we expanded our object detection comparison to
include four recent SOTA methods: UniMamba [1], FSHNet [2], ScatterFormer [3], and LION [4]
(Tables 1 & 5). Our scaled-up backbone, with comparable model size to these methods,
achieves higher mAP and NDS on nuScenes, higher mAPH on Waymo, and is only slightly below
FSHNet in Waymo mAP. Notably, while these references are designed specifically for detection,
our backbone performs strongly across both detection and segmentation, demonstrating its
versatility as a more general 3D representation learner.

**3. Runtime and model-size analysis added**

Per reviewer nrxi suggestion, we added a computational evaluation comparing parameters and
runtime against strong baselines, in Table 5 Section 5.4. The results show that our backbone
reaches state-of-the-art accuracy with comparable efficiency to methods such as UniMamba [1]
and FSHNet [2], while outperforming them in accuracy. Although some approaches, such as
ScatterFormer [3], run faster, our model achieves higher accuracy (+1.3 mAPH on Waymo, +0.6
mAP on nuScenes).

Reviewer AzB2 noted that these improvements over ScatterFormer may appear incremental.
However, both Waymo and nuScenes are highly saturated benchmarks, where top methods
typically improve by less than one point. Recent publications in top-tier conferences (e.g., LION, UniMamba, Voxel
Mamba, FSHNet, SAFDNet) report gains of similar or even smaller scale. Within this context, our
improvements are meaningful, competitive, and aligned with current progress in 3D
perception.

**4. Visualization of Uniform Dilated Grouping**

Following reviewer mcbN, we added qualitative visualizations illustrating the core mechanism
of Uniform Dilated Grouping (Appendix I). These examples show how our method normalizes
geometric density across range on real point cloud data.

**5. Qualitative detection results**

Per reviewer RTGD’s suggestion, we added detection visualization comparing our predictions to
ground truth (Appendix J).

**6. Limitations & future work included**

We added a dedicated Limitations and Future Work section (Appendix H), acknowledging that
our range-equalization design is less impactful in short-range indoor segmentation, where
performance remains below state-of-the-art, and outlining future extensions to broader long-range understanding tasks such as classification.


[1] X Jin et al. "UniMamba: Unified Spatial-Channel Representation Learning with Group-
Efficient Mamba for LiDAR-based 3D Object Detection.", CVPR 2025.
[2] S Liu et al. "FSHNet: Fully Sparse Hybrid Network for 3D Object Detection.", CVPR 2025.
[3] He, C. et al. “ScatterFormer: Efficient Voxel Transformer with Scattered Linear Attention.”,
ECCV 2024.
[4] Z Liu et al. "LION: Linear Group RNN for 3D Object Detection in Point Clouds.", NeurIPS 2024.

---

### Meta-Review · Area_Chair_RBL5 · 2025-12-11

**Summary:**

The AC has carefully examined the paper, the reviews, the rebuttal, the discussions, and the revisions. The reviewers, as well as the AC, acknowledge the contributions of the proposed method, particularly the novel and well-structured design of the LR-DUT module. However, several main concerns remain, which prevent recommending acceptance at this time:

1. Although the method is described as “an efficient architecture,” the newly added results show that it uses a similar number of parameters and has comparable inference latency to existing Transformer/Mamba/CNN approaches, and is in fact slower than HEDNet and ScatterFormer, which achieve similar accuracy as the proposed method.

2. Qualitative comparisons with existing methods are still absent from the current manuscript. Such comparisons should be included in the main paper, with additional qualitative results provided in the supplementary material where possible. The authors may want to further adjust the layout and content distribution of the manuscript.

3. The title is somewhat generic and should be refined to more specifically reflect the proposed technique and its target tasks. In its current form, readers may either overlook the paper or form overly broad expectations that are not fully aligned with the actual scope of the work. If the authors would like to highlight the generalization ability of this method, more point cloud related tasks shall be tested and compared. The main text could be further polished, including more careful cross-references to figures and equations, and more consistent terminology throughout the paper.

Overall, while the paper has clear merits, it is not yet ready for publication at ICLR 2026 given these key issues. The AC encourages the authors to address these points and submit a revised version to a related venue next time.

**Reviewer Concerns:**

Please refer to the summary above.

**Reviewer Scores:**

Reviewer nrxi: 4 → 6 ("I will raise my score to 6.")
Reviewer mcbN: 6 (no reply)
Reviewer AzB2: 4 (replied but maintained)
Reviewer RTGD: 4 (replied but still concerned)

---

### Decision · Program_Chairs · 2026-01-26

Reject